# GRAM domain proteins specialize functionally distinct ER-PM contact sites in human cells

Marina Besprozvannaya[1], Eamonn Dickson[2], Hao Li[3], Kenneth S Ginburg[4], Donald M Bers[4], Johan Auwerx[3], Jodi Nunnari[1]*

[1]Department of Molecular and Cellular Biology, University of California, Davis, Davis, United States; [2]Department of Physiology and Membrane Biology, School of Medicine, University of California, Davis, Davis, United States; [3]Laboratory of Integrative and Systems Physiology, EPFL, Lausanne, Switzerland; [4]Department of Pharmacology, University of California, Davis, Davis, United States

**Abstract** Endoplasmic reticulum (ER) membrane contact sites (MCSs) are crucial regulatory hubs in cells, playing roles in signaling, organelle dynamics, and ion and lipid homeostasis. Previous work demonstrated that the highly conserved yeast Ltc/Lam sterol transporters localize and function at ER MCSs. Our analysis of the human family members, GRAMD1a and GRAMD2a, demonstrates that they are ER-PM MCS proteins, which mark separate regions of the plasma membrane (PM) and perform distinct functions in vivo. GRAMD2a, but not GRAMD1a, co-localizes with the E-Syt2/3 tethers at ER-PM contacts in a PIP lipid-dependent manner and pre-marks the subset of PI(4,5)P2-enriched ER-PM MCSs utilized for STIM1 recruitment. Data from an analysis of cells lacking GRAMD2a suggest that it is an organizer of ER-PM MCSs with pleiotropic functions including calcium homeostasis. Thus, our data demonstrate the existence of multiple ER-PM domains in human cells that are functionally specialized by GRAM-domain containing proteins.
DOI: https://doi.org/10.7554/eLife.31019.001

*For correspondence:
jmnunnari@ucdavis.edu

## Introduction

The endoplasmic reticulum (ER) in eukaryotic cells is a vast and distributed intracellular network of membranous tubes and sheets with essential roles in ion and lipid homeostasis. It exerts many of its functions through intimate membrane contact sites (MCSs) with other organelles (*Prinz, 2014*). MCSs are regions where organelles typically come within 10 to 30 nm of one another and are thought to be specialized 'microdomains' that selectively localize lipid and protein effectors that function in signaling pathways and organelle dynamics (*Eisenberg-Bord et al., 2016*; *Murley and Nunnari, 2016*; *Saheki and De Camilli, 2017a*).

MCSs between the ER and the plasma membrane (ER-PM) are a highly conserved feature of eukaryotic cells and have emerged as key regulators of intracellular $Ca^{2+}$ dynamics (*Chang and Liou, 2016*; *Dickson et al., 2016a*; *Eisenberg-Bord et al., 2016*; *Henne et al., 2015*; *Saheki and De Camilli, 2017a*; *Stefan et al., 2013*). In particular, in metazoan cells the store operated calcium entry pathway (SOCE) is critical for maintaining cellular $Ca^{2+}$ homeostasis and is activated by the depletion of $Ca^{2+}$ stores in the ER, which triggers extracellular $Ca^{2+}$ influx through the PM at ER-PM contact sites to refill ER lumen $Ca^{2+}$ stores (*Carrasco and Meyer, 2011*; *Lewis, 2011*). Stromal-Interacting Molecule 1 (STIM1) protein is an integral ER protein that regulates SOCE in response to luminal ER $Ca^{2+}$. A decrease in intraluminal ER $Ca^{2+}$ is sensed by STIM1, which undergoes conformational changes that expose domains that promote oligomerization and transport/targeting to pre-existing phosphatidylinositol 4,5-biphosphate (PI(4,5)P2) enriched ER-PM MCSs (*Carrasco and Meyer, 2011*;

*Várnai et al., 2007*; *Zhang et al., 2005*). At PI(4,5)P2-enriched ER-PM contacts, STIM1 directly recruits and activates PM Orai1 $Ca^{2+}$ channels, which replenish ER $Ca^{2+}$ (*Carrasco and Meyer, 2011*; *Chang and Liou, 2016*; *Maléth et al., 2014*; *Sharma et al., 2013*).

Highly conserved ER-PM tethers have been described and include the yeast tricalbins/human extended synaptotagmins (E-Syt) (*Giordano et al., 2013*; *Manford et al., 2012*; *Saheki and De Camilli, 2017b*). Mammalian E-Syt1/2/3 are anchored to the ER via an N-terminal hydrophobic hairpin and possess repeating cytosolic C2 domains, which directly mediate contact between the ER and the PM via interactions with PM-enriched PI(4,5)P2 lipids (*Giordano et al., 2013*). In addition, they contain S̲ynaptotagmin-like, M̲itochondrial and lipid-binding P̲rotein (SMP) domains, which mediate their homo- and hetero-oligomerization and facilitate glycerolipid transport – a function implicated during PLC activation in the transport and recycling of diacylglycerol from the PM to the ER in mammalian cells (*Fernández-Busnadiego et al., 2015*; *Saheki et al., 2016*; *Schauder et al., 2014*). E-Syt2/3 tether the ER to the PM at resting $Ca^{2+}$ levels, while E-Sty1 is recruited to ER-PM in a manner dependent on conditions of high intracellular $Ca^{2+}$, as observed during SOCE where E-Syt1 stabilizes and expands $Ca^{2+}$-specific microdomains (*Chang et al., 2013*; *Maléth et al., 2014*). During SOCE, STIM1 and Orai1 are recruited to a subset of E-Syts-containing ER-PM contact sites; however, the E-Syts are not required for any events in the SOCE pathway (*Giordano et al., 2013*). In addition, cells lacking all three known E-Syts (1/2/3) still possess 50% of the ER-PM contact area relative to wild type cells (*Giordano et al., 2013*; *Saheki et al., 2016*). These observations suggest that additional ER-PM tethers exist and potentially function to specialize ER-PM contacts (*Saheki and De Camilli, 2017a*).

A highly conserved family of ER membrane Ltc/Lam proteins was recently described (*Elbaz-Alon et al., 2015*; *Gatta et al., 2015*; *Murley et al., 2015*) which, within yeast cells, localize to MCSs where they facilitate the intermembrane transfer of sterol lipids via VaST domains (*Figure 1A*) (*Elbaz-Alon et al., 2015*; *Gatta et al., 2015*; *Murley et al., 2015*). In yeast, Ltc1/Lam6 localizes to ER-mitochondria and ER-lysosome/vacuole contacts via physical interactions with the protein partners, Tom70/71 and Vac8, respectively (*Gatta et al., 2015*; *Murley et al., 2015*). Ltc3/Lam4 and Ltc4/Ysp2 paralogs are localized to ER-PM MCSs, where they have been implicated in the retrograde transport of sterols from the PM to the ER and in regulation of PM TORC2 activity to coordinate sterol and sphingolipid homeostasis (*Gatta et al., 2015*; *Murley et al., 2017*). Here we examined the functions of the uncharacterized human Ltc/Lam protein orthologs, GRAMD1a and GRAMD2a. Our analysis indicates that GRAMD1a and GRAMD2a are ER-PM tethers that mark and define functionally distinct regions of the PM.

## Results

### GRAM domain-dependent targeting of GRAMD1a and GRAMD2a to ER-PM MCSs

Yeast Ltc/Lam proteins possess a common domain structure that includes an N-terminal unstructured region, a PH-like GRAM domain, which in the context of MTMR2 binds PIP lipids; one or two START-like VASt domains, which bind and facilitate sterol transport; and a C-terminal transmembrane domain (TMD) helix, which anchors them into ER membrane (*Figure 1A*) (*Begley et al., 2003*; *Berger et al., 2003*; *Gatta et al., 2015*; *Murley et al., 2015*). Using phylogenetic analysis and structure prediction programs, we identified the uncharacterized GRAM-domain containing proteins, GRAMD1a-c, GRAMD2, and GRAMD3 as human Ltc orthologs. Our analysis indicates that the human family is characterized by the presence of a related GRAM domain as only GRAMD1a-c share the yeast canonical domain structure (*Figure 1A and B*; *Figure 1—figure supplement 1A*). GRAMD2 and GRAMD3 lack a VASt domain and only possess a related GRAM and ER-anchoring TM domain (*Figure 1A*). Accordingly, with permission from the HUGO Gene Nomenclature Committee, we renamed GRAMD2 and GRAMD3 as GRAMD2a and GRAMD2b, respectively.

To gain insight into function, we examined the intracellular localization of transiently expressed fluorescently labeled versions of the human Ltc/Lam orthologs in mammalian cells using spinning disk microscopy. Both GRAMD1a-eGFP and GRAMD2a-eGFP were observed in focal structures at the periphery of Cos7 cells, suggesting that, similar to previously characterized yeast Ltc3/4 (Lam4/Ysp2) proteins, they localize to ER-PM contact sites (*Figure 1C and D*). To test this, we examined

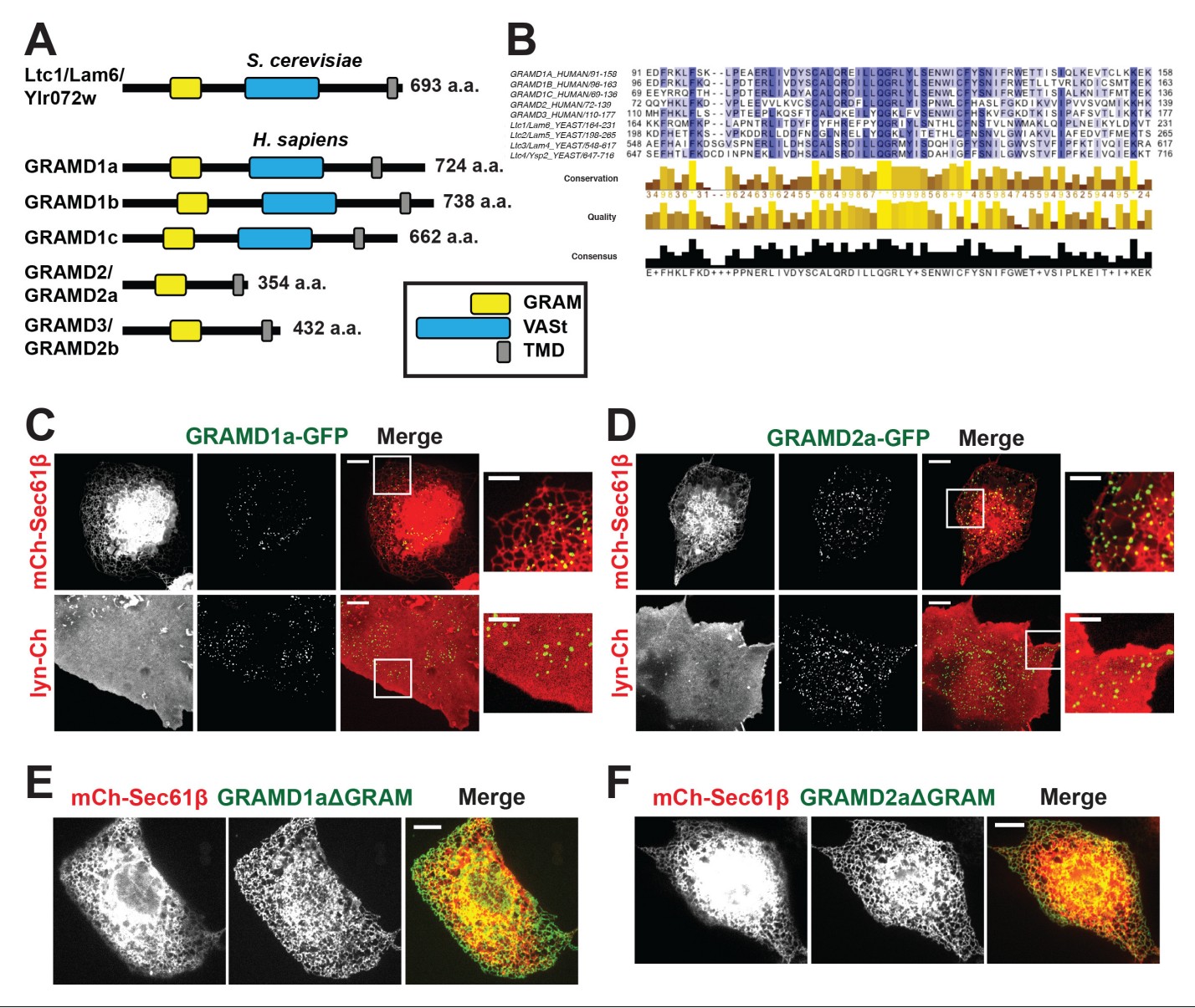

**Figure 1.** Defining a new family of human ER MCS proteins. (**A**) Human proteins GRAMD1a-c and GRAMD2a and b are members of a protein family with similarity to the yeast Ltc1/Lam proteins, which all possess an unstructured N-terminus, a GRAM domain, and an ER-anchoring hydrophobic transmembrane domain. GRAMD1a-c contain an additional predicted sterol transport START-like VaST domain, similar to yeast family member Ltc1/Lam6. (**B**) Alignment of conserved GRAM domains from yeast and human family members. (**C–D**) Localization of GRAMD1a-eGFP (**C**) and GRAMD2a-eGFP (**D**) in COS7 mammalian cells relative to ER-marker mCherry-Sec61β and PM-marker lyn-mCherry. (**E–F**) Examination of localization of GRAMD1aΔGRAM (**E**) and GRAMD2aΔGRAM (**F**) with mCherry-Sec61β. Representative images shown from at least 17 cells that were obtained from three biological replicates. Scale bar 10 μm in full images and 5 μm in insets (the same dimensions are maintained for all subsequent figures).
DOI: https://doi.org/10.7554/eLife.31019.002

The following video and figure supplement are available for figure 1:

**Figure supplement 1.** GRAMD1a and GRAMD2a localize to ER-PM contact sites.
DOI: https://doi.org/10.7554/eLife.31019.003
**Figure 1—video 1.** GRAMD1a-eGFP with mCherry-Sec61β and lyn-mCherry Z-stack, sample images are displayed in *Figure 1C*
DOI: https://doi.org/10.7554/eLife.31019.004
**Figure 1—video 2.** GRAMD2a-eGFP with mCherry-Sec61β and lyn-mCherry Z-stack, sample images are displayed in *Figure 1D*
DOI: https://doi.org/10.7554/eLife.31019.005
**Figure 1—video 3.** 3D rendering of GRAMD1a-eGFP with mCherry-Sec61β and lyn-mCherry: orthogonal view, sample images displayed in *Figure 1—figure supplement 1B*
*Figure 1 continued on next page*

*Figure 1 continued*

DOI: https://doi.org/10.7554/eLife.31019.006

**Figure 1—video 4.** 3D rendering of GRAMD2a-eGFP with mCherry-Sec61β and lyn-mCherry: orthogonal view, sample images displayed in *Figure 1—figure supplement 1C*

DOI: https://doi.org/10.7554/eLife.31019.007

their localization relative to an ER marker, mCherry-Sec61β, or PM marker, lyn-mCherry. Consistently, analysis of Z stack images and Z-stack reconstructions of cells indicate that peripheral GRAMD1a-eGFP and GRAMD2a-eGFP foci co-localized with ER and PM (*Figure 1C and D* and *Figure 1—figure supplement 1B and C*; *Figure 1—video 1*, *2*, *3*, *4*). Additionally, we used total internal reflection (TIRF) microscopy, which allows selective illumination within ~100 nm of the PM, to examine cells expressing either GRAMD1a-eGFP or GRAMD2a-eGFP and both BFP-Sec61β and lyn-mCherry (*Figure 1—figure supplement 1D and E*). Line scans of individual GRAMD1a or GRAMD2a puncta revealed that GRAMD1a and GRAMD2a labeled focal structures co-localized with cortical ER coincident with the PM (*Figure 1—figure supplement 1D and E*). Thus, our cytological data indicate that GRAMD1a and GRAMD2a localize to ER-PM MCSs.

We examined the molecular basis for GRAMD1a and GRAMD2a targeting to ER-PM MCSs. Previously, it was shown that the GRAM domain of yeast Ltc1 is required for mitochondrial localization of Ltc1 (*Murley et al., 2015*), suggesting that the GRAM domain is a critical determinant of MCS localization. Consistent with this model, variants of GRAMD1a and GRAMD2a lacking their GRAM domain (GRAMD1aΔGRAM-eGFP and GRAMD2aΔGRAM-eGFP, respectively) exhibited diffuse ER localization and were not observed to localize in focal structures at ER-PM MCSs (*Figure 1E and F*, respectively). These data indicate that GRAMD1a and GRAMD2a are targeted to the PM via their GRAM domains and further suggest that this targeting mechanism is a general feature of the family.

## GRAMD1a and GRAMD2a localize to distinct ER-PM MCSs

To further characterize GRAMD1a and GRAMD2a cortical structures, we examined their relationship to each other. In cells co-expressing both GRAMD2a-mCherry and GRAMD1a-eGFP, no co-localization of GRAMD2a and GRAMD1a labeled foci was observed as only ~8% of total GRAMD2a-eGFP fluorescence pixels overlapped with GRAMD1a-eGFP fluorescent pixels and visa versa (*Figure 2A*). These data suggest that GRAMD1a and GRAMD2a mark distinct ER-PM contact sites. To test this idea, we examined the relationship of GRAMD2a and GRAMD1a marked cortical foci to the localization of E-Syts2/3, which are well characterized ER-PM tethers that hetero-oligomerize and localize to ER-PM contacts via interaction with PM PI(4,5)P$_2$ lipid domains (*Figure 2B and C*) (*Fernández-Busnadiego et al., 2015*; *Giordano et al., 2013*). Analysis of cells expressing GRAMD2a-eGFP and either mCherry-E-Syt2 or E-Syt3-mCherry revealed that a majority of the GRAMD2a-eGFP fluorescence signal co-localized with mCherry-E-Syt2 and E-Syt3-mCherry signals (75.9 ± 2.5% and 85.7 ± 3.3% respectively; *Figure 2B and D*). Consistently, line-scan analysis indicated that GRAMD2a-labeled foci were co-localized with E-Syt3-labeled regions and also with a significant fraction of E-Syt2-labeled regions at the cell cortex (*Figure 2B*, bottom panel). Additionally, TIRF imaging of cells expressing ER marker BFP-Sec61β with GRAMD2a-eGFP and either mCherry-E-Syt2 or E-Syt3-mCherry demonstrated that co-localized regions of GRAMD2a-eGFP with either mCherry-E-Syt2 and E-Syt3-mCherry corresponded to regions of cortical ER at the cell periphery, consistent with a shared localization at ER-PM contacts (*Figure 2—figure supplement 1A and B*). Although GRAMD2a extensively co-localized with E-Syt2 and E-Syt3 at ER-PM MCSs, a similar GRAMD2a-eGFP punctate localization pattern at cortical ER in both wildtype and E-Syt1/2/3 triple knock out HeLa cells was observed (*Figure 2—figure supplement 1C*). These data indicate that GRAMD2a is targeted to ER-PM MCSs independent of E-Syts. In contrast to GRAMD2a, GRAMD1a-eGFP fluorescence was not significantly co-localized with either mCherry-E-Syt2 or E-Syt3-mCherry fluorescence at the cell cortex (8.9 ± 1.8% and 8.9 ± 1.0% respectively and *Figure 2C and D*; *Figure 2—figure supplement 1D and E*). Together our data demonstrate that GRAMD2a and GRAMD1a localize to and distinguish distinct ER-PM MCSs.

Localization of GRAMD2a and GRAMD1a to distinct ER-PM domains suggests that they possess distinct physiological functions. To infer potential roles of these genes, we performed gene set enrichment analysis (GSEA) using their expression levels as input phenotypes to find gene sets that

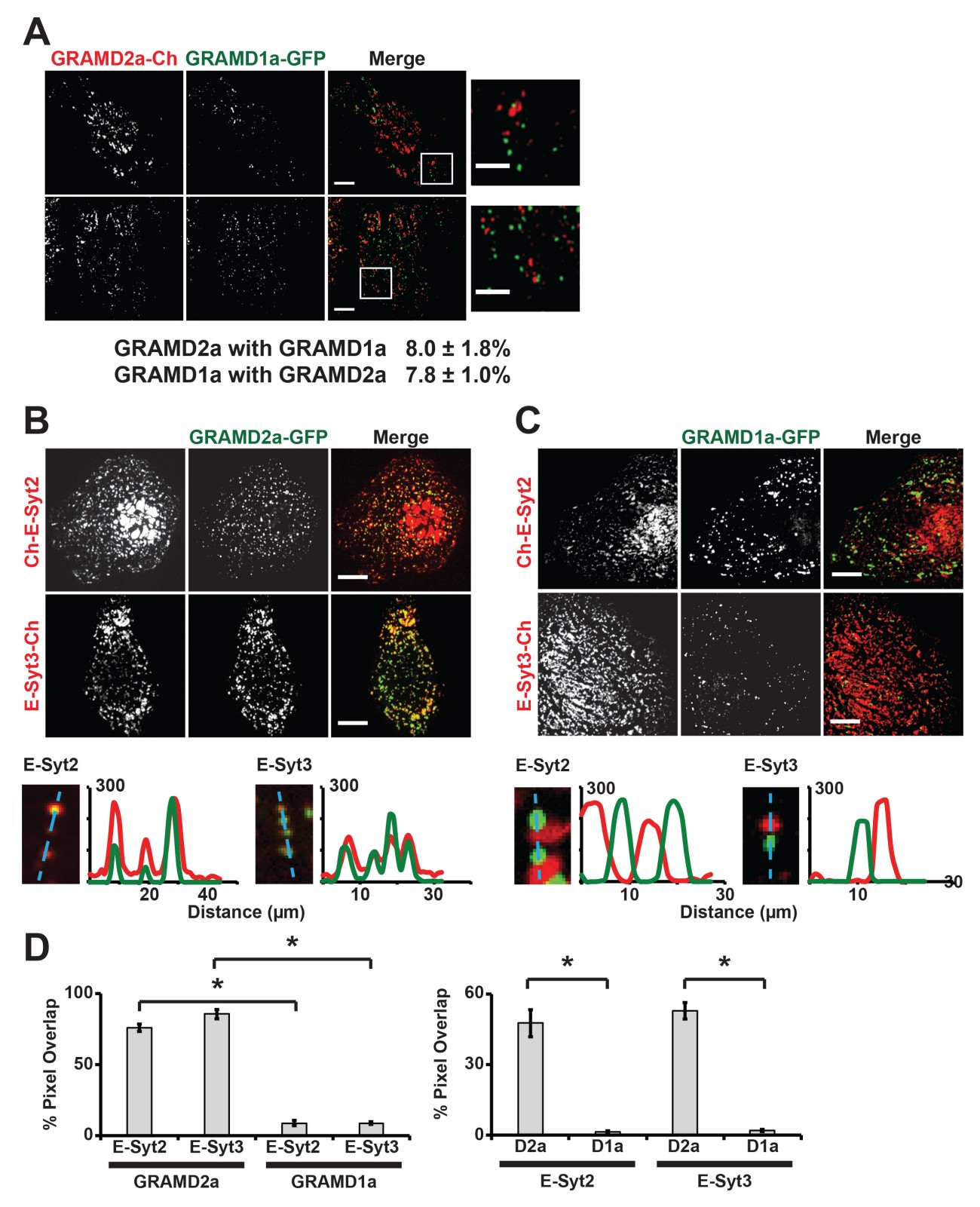

**Figure 2.** GRAMD2a and GRAMD1a mark distinct ER-PM contact sites. (**A**) Analysis of the relative localization of GRAMD1a-eGFP and GRAMD2a-mCherry in Cos7 cells. Two sample cells are shown. Bottom panel is quantification of the amount of overlapping total fluorescence signal pixels from GRAMD1a-eGFP and GRAMD2a-mCherry. Representative images of two different cells, that were imaged on two different days shown. 14 cells imaged and analyzed from three biological replicates. (**B–C**) Localization of GRAMD2a-eGFP (**B**) or GRAMD1a-eGFP (**D**) relative to mCherry-labeled ER-PM
*Figure 2 continued on next page*

*Figure 2 continued*

tethers E-Syt2 or E-Syt3 in Cos7 cells. Bottom panels are line scan analysis of individual GRAMD2a-eGFP (B) or GRAMD1a-eGFP (C) foci. Y-axis of line scans are arbitrary fluorescence units. (D) Quantification of co-localization of GRAMD2a or GRAMD1a with E-Syt2 and E-Syt3 as % of total fluorescence pixels of GRAMD1a or GRAMD2a (top panel) or ESyt2 or ESyt3 (bottom panel). Standard Error shown, *$p<10^{-2}$ from two-tailed t-test. Specifically, GRAMD2a with E-Syt2 compared to GRAMD1a with E-Syt2, $p=8.97 \times 10^{-12}$; GRAMD2a with E-Syt3 compared to GRAMD1a with E-Syt3, $p=3.40 \times 10^{-17}$; E-Syt2 with GRAMD2a compared to E-Syt2 with GRAMD1a, $p=9.56 \times 10^{-7}$; E-Syt3 with GRAMD2a compared to E-Syt3 with GRAMD1a, $p=3.14 \times 10^{-13}$. Representative images shown from at least 16 cells that were obtained from three biological replicates. Pixel co-localization analysis was performed on all cells.

DOI: https://doi.org/10.7554/eLife.31019.008

The following source data and figure supplement are available for figure 2:

**Source data 1.** *Figure 2D* Co-localization analysis: Top table is pixel overlap of GRAMD2a and GRAMD1a with E-Syt2 and E-Syt3; standard Error shown.
DOI: https://doi.org/10.7554/eLife.31019.010
**Figure supplement 1.** Relationship of GRAMD1a and GRAMD2a to E-Syt2/3.
DOI: https://doi.org/10.7554/eLife.31019.009

co-expressed with GRAMD genes (*Subramanian et al., 2005*) in transcriptome data from liver samples of human (*Schadt et al., 2008*) and mouse populations (*Wu et al., 2014b*). Consistent with their different intracellular localizations, we found that *GRAMD1A* and *GRAMD2A* transcripts exhibited quite diverse correlated pathways (*Figure 3A* and *Figure 3—figure supplement 1A*), suggesting their distinct functions. Specifically, *GRAMD2a/Gramd2a* exhibited robust positive correlations with genes involved in lipid metabolism in human and mouse populations, while *GRAMD1a/Gramd1a* showed opposite correlation patterns (*Figure 3A*, green gene-sets; *Figure 3B*, upper panel; *Figure 3C*, left panel; and *Figure 3—figure supplement 1B*, left panel). These observations indicate that GRAMD1a and GRAMD2a possess distinct functions in mammals in vivo, consistent with our cellular data demonstrating that they localize to distinct ER-PM contacts.

## GRAMD2a targeting to PM is dependent on PI(4,5)P2

PI(4,5)P2 is highly enriched in ER-PM contact sites and mediates the localization of E-Syt2/3 to ER-PM MCSs (*Chang and Liou, 2016*; *Dickson et al., 2016a*; *Saheki and De Camilli, 2017a*). Thus, given the co-localization of GRAMD2a with E-Syt2 and E-Syt3, we asked if GRAMD2a targets to PM MCSs in cells in a manner dependent on PIP lipids. To test this, PI(4,5)P2 was depleted from the PM by stimulating phosphoinositide-specific phospholipase C (PLC) using the muscarinic agonist oxotermorine-M (OxoM) in cells transfected with the muscarinic acetylcholine receptor (M1R) (*Dickson et al., 2016b*; *Giordano et al., 2013*). We imaged cells overexpressing untagged M1R and monitored PM PI(4,5)P2 using the specific fluorescent biosensor CFP-PH-PLC$_{\delta 1}$. Following addition of Oxo-M, PM associated CFP-PH-PLC$_{\delta 1}$ and GRAMD2a-mCherry coordinately and significantly decreased. Following removal of Oxo-M from the media both PM associated CFP-PH-PLC$_{\delta 1}$ and GRAMD2a-mCherry increased but with different kinetics; PM associated GRAMD2a-mCherry fluorescence more rapidly recovered as compared the PI(4,5)P2 selective marker, CFP-PH-PLC$_{\delta 1}$ (*Figure 4A and B*, top panel). In contrast, GRAMD1a-mCherry fluorescence remained associated with the cortex following addition of Oxo-M, although PM-associated CFP-PH-PLC$_{\delta 1}$ fluorescence decreased (*Figure 4A and B*, bottom panel). These observations suggest that targeting of GRAMD2a to ER-PM contacts is at least in part dependent on PI(4,5)P2, while GRAMD1a targeting is PI(4,5)P2-independent, further substantiating that GRAMD2a and GRAMD1a localize to distinct ER-PM domains in human cells.

To more directly test the idea that GRAMD2a PM targeting is dependent on PI(4,5)P2, we expressed and purified the predicted GRAMD2a cytosolic domain (amino acids 1–298, 6xHis-GRAMD2aΔTMD) and performed liposome-binding assays (*Figure 4—figure supplement 1A*). Western blotting analysis of liposome binding assays revealed that GRAMD2aΔTM significantly and specifically associated with liposomes containing either PI(4)P and PI(4,5)P2 in a concentration-dependent manner (*Figure 4C*), with an apparent higher selectivity for PI(4,5)P2 versus PI(4)P (*Figure 4D*). The ability of GRAMD2a to bind PI(4)P in addition to PI(4,5)P2 in vitro suggests that it may bind PI(4)P, a precursor of PI(4,5)P2 in the PM, in cells and thus could be the basis for the relatively rapid kinetics of PM binding of GRAMD2a after removal of Oxo-M after PIP depletion in cells as compared to the PI(4,5)P2 marker (*Figure 4A*) (*Dickson et al., 2016a*). These biochemical observations are consistent

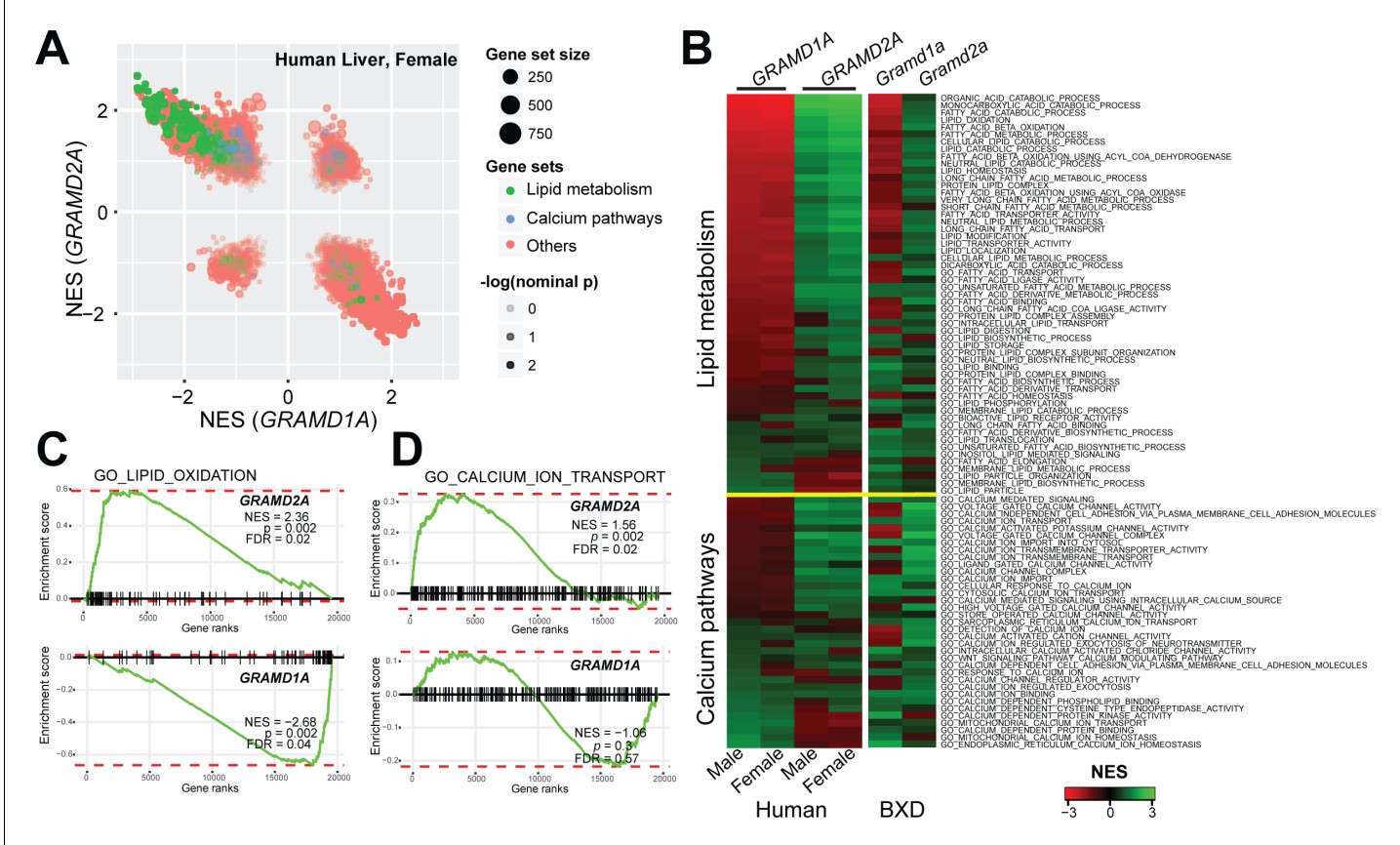

**Figure 3.** Gene set enrichment analysis of GRAMD1a and GRAMD2a indicated distinct physiological functions. (**A**) Comparison of enrichment results between *GRAMD1a* and *GRAMD2a* in transcriptome data of liver samples from 193 female human individuals. Normalized enrichment score (NES) of *GRAMD1a* and *GRAMD2a* are used to compare the GO pathway enrichment of these two genes in lipid metabolism and $Ca^{2+}$ signaling gene sets highlighted in green and blue, respectively. Dot size represents the number of genes, and transparency of the dot indicates the significance (-$\log_{10}$(nominal *p* value)) of the enrichment of the two transcripts for the gene set. (**B**) Heat-map showing the enrichment of *GRAMD1a* and *GRAMD2a* in genes involved in lipid metabolism and $Ca^{2+}$ signaling in liver samples from human male and female individuals, as well as from males of the BXD mouse genetic reference population. (**C–D**) Enrichment plot of *GRAMD1a* and *GRAMD2a* in human liver samples from female individuals shows their distinct physiological functions in lipid metabolism (**C**) and $Ca^{2+}$ signaling pathways (**D**). FDR, false discovery rate.

DOI: https://doi.org/10.7554/eLife.31019.011

The following figure supplement is available for figure 3:

**Figure supplement 1.** Gene set enrichment analysis of GRAMD1a and GRAMD2a in males.

DOI: https://doi.org/10.7554/eLife.31019.012

with our cytological data and indicate that GRAMD2a is targeted to the PM by directly interacting with PIP lipids. Thus, with data indicating that the GRAM domain is required for PM targeting (*Figure 1D*), GRAMD2a functions as a ER anchored-PM tether whose PM binding is mediated via a GRAM domain–PIP-lipid interaction.

## GRAMD2a pre-marks a subset of ER-PM contacts used for STIM1 recruitment

To gain insight into the cellular function of GRAMD2a, we asked whether its over-expression affected the extent or predominance of ER-PM contacts as measured by amount of cortical ER in cells. Increasing the amount of transfected GRAMD2a-mCherry plasmid DNA increased the amount of GRAMD2a expression (*Figure 5—figure supplement 1A*) and resulted in a significant increase in the area of cortical ER, marked by GFP-Sec61β, which co-localized with GRAMD2a (compare *Figure 1D* with *Figure 5A*, top panel). The GRAMD2a expression-dependent cortical ER expansion

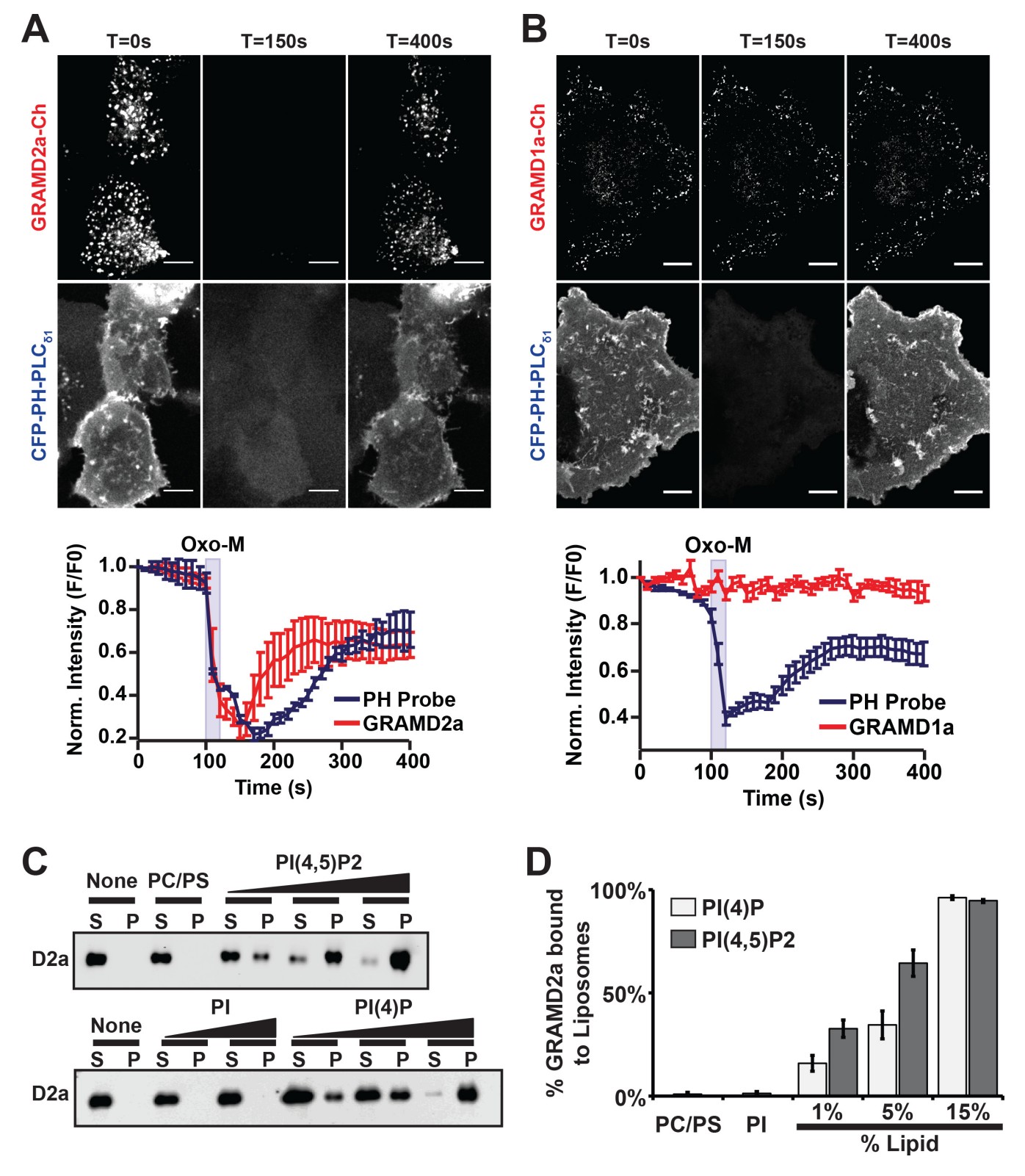

**Figure 4.** GRAMD2a is targeted to the PM via PI(4,5)P2. (A–B) Examination of the PI(4,5)P2-dependent behavior of PM-associated GRAMD2a-eGFP (A) and GRAMD1a-eGFP (B). PM PI(4,5)P2 was reversibly depleted by 10 s addition (middle panel) and removal (left panel) of 10 μM of the muscarinic agonist oxotremorineM (OxoM) as monitored by the PI(4,5)P2 marker CFP-PH-PLC$_{\delta 1}$. Normalized intensity of fluorescent proteins used in PI(4,5)P2 depletion experiments shown in lower panels of (A and B). Representative images shown from at least 12 cells that were obtained from two biological

*Figure 4 continued on next page*

*Figure 4 continued*

replicates. Fluorescence intensity dynamics were analyzed for all cells. (C) Western Blot analysis of centrifugation-based liposome binding assays with recombinant His6-GRAMD2aΔTM and liposomes of different composition. Control PM-like liposomes: 85% PC: 15% PS; PM-like liposomes with PI: 85% PC: 10% PS: 5% PI or 85% PC: 15% PI; PM-like liposomes with PI(4)P: 85% PC: 14% PS: 1% PI(4)P, 85% PC: 10% PS: 5% PI(4)P, or 85% PC: 15% PI(4)P; PM-like liposomes with PI(4,5)P2: 85% PC: 14% PS: 1% PI(4,5)P2, 85% PC: 10% PS: 5% PI(4,5)P2, or 85% PC: 15% PI(4,5)P2. S and P indicate supernatant and pellet, respectively. (D) Quantification of liposome binding experiments, n = 4 (biological replicates), Standard Error shown.
DOI: https://doi.org/10.7554/eLife.31019.013

The following source data and figure supplement are available for figure 4:

**Source data 1.** *Figure 4D* Quantification of liposome binding assays with recombinant His6-GRAMD2aΔTM: liposome assays were repeated four times; standard Error shown.
DOI: https://doi.org/10.7554/eLife.31019.015

**Figure supplement 1.** Schematic depiction of liposome binding assays.
DOI: https://doi.org/10.7554/eLife.31019.014

was similar to that previously described for ER-PM micro-domains or plasters linked to SOCE. Indeed, expression of mCherry-STIM1, an integral ER protein that regulates SOCE in response to decreased ER Ca$^{2+}$ levels, also resulted in the expansion of cortical ER, consistent with published data (*Figure 5A*, bottom panel) (*Liou et al., 2007*; *Orci et al., 2009*). These observations suggest that GRAMD2a marks SOCE-specific ER-PM MCSs. To test the link between GRAMD2a and Ca$^{2+}$ signaling, we performed GSEA on *GRAMD2a* and *GRAMD1a* with calcium signaling related pathways (*Subramanian et al., 2005*). *GRAMD2a* showed strong positive correlations with Ca$^{2+}$ signaling pathways across human and mouse datasets, while *GRAMD1*a did not correlate consistently with Ca$^{2+}$ signaling (*Figure 3A*, blue gene sets; *Figure 3B*, lower panel; *Figure 3C*, right panel; and *Figure 3—figure supplement 1B*, right panel). Indeed, regardless of the amount transfected, GRAMD1a-mCherry retained a similar cortical punctate area and distribution in cells (*Figure 5—figure supplement 1A*; compare *Figure 1C* to *Figure 5—figure supplement 1B*). These results suggest a unique functional role of GRAMD2a in Ca$^{2+}$ signaling.

To directly test whether GRAMD2a is spatially linked to SOCE, we examined whether it co-localized with STIM1 upon ER Ca$^{2+}$ store depletion mediated by thapsigargin (TG), a non-competitive inhibitor of the sarcolemma ER Ca$^{2+}$ ATPase (*Giordano et al., 2013*). We implemented a prototypic Ca2+ add back experiment to monitor the SOCE response in which TG-induces Ca2+ release from internal stores and consequent Ca2+ influx across the PM (*Chen et al., 2017*). As previously described, in the absence of TG, mCherry-STIM1 was diffusely distributed throughout the ER (*Grigoriev et al., 2008*). Upon TG addition, mCherry-STIM1 rapidly re-distributed to discrete regions marked by GRAMD2a-eGFP at the cell cortex, which, based on previous observations, represent ER-PM contact sites (*Figure 5B*, top panel two panels, and *Figure 5—video 1* and *2*) (*Liou et al., 2007*; *Zhang et al., 2005*). Consistently, after TG treatment total cortical fluorescence of mCherry-STIM1 increased significantly whereas the fluorescence intensity of cortical GRAMD2a-eGFP remained constant (*Figure 5B*, lower panel). To assess whether STIM1 and GRAMD2a co-localized at the cell cortex, we performed line scans of fluorescent intensity across GRAMD2a-eGFP foci before and after addition of TG. As expected, in the absence of TG, line scans indicated no significant co-localization of mCherry-STIM1 fluorescence with cortical GRAMD2a-eGFP foci (*Figure 5C* and *Figure 5—figure supplement 1C*). Initially after TG addition, line scans indicated significant co-localization of mCherry-STIM1 fluorescence with GRAMD2a foci (*Figure 5C* and *Figure 5—figure supplement 1C*, T50-T150) and, subsequently, regions of mCherry-STIM1 cortical fluorescence resolved from, but remained spatially linked to GRAMD2a-eGFP foci (*Figure 5C* and *Figure 5—figure supplement 1C*, T150-T250). Consistent with our qualitative line scan analysis, the percentage of overlapping fluorescence pixels of GRAMD2a-eGFP with mCherry-STIM1 significantly increased following TG addition (*Figure 5—source data 1*). The percentage of overlapping fluorescence pixels of mCherry-STIM1 with GRAMD2a-eGFP initially increased following TG addition, but as more mCherry-STIM1 translocated to the PM, the percentage of overlapping mCherry-fluorescence pixels of mCherry-STIM1 with GRAMD2a-eGFP decreased (*Figure 5—source data 1*). Although areas of STIM1 and GRAMD2a fluorescence spatially resolved at later time points of TG treatment, line scan analysis the ER using ER marker BFP-Sec61β revealed that STIM1 and GRAMD2a remained co-localized to a shared ER-PM contact site (*Figure 5D*). In contrast to GRAMD2a, STIM1 did not

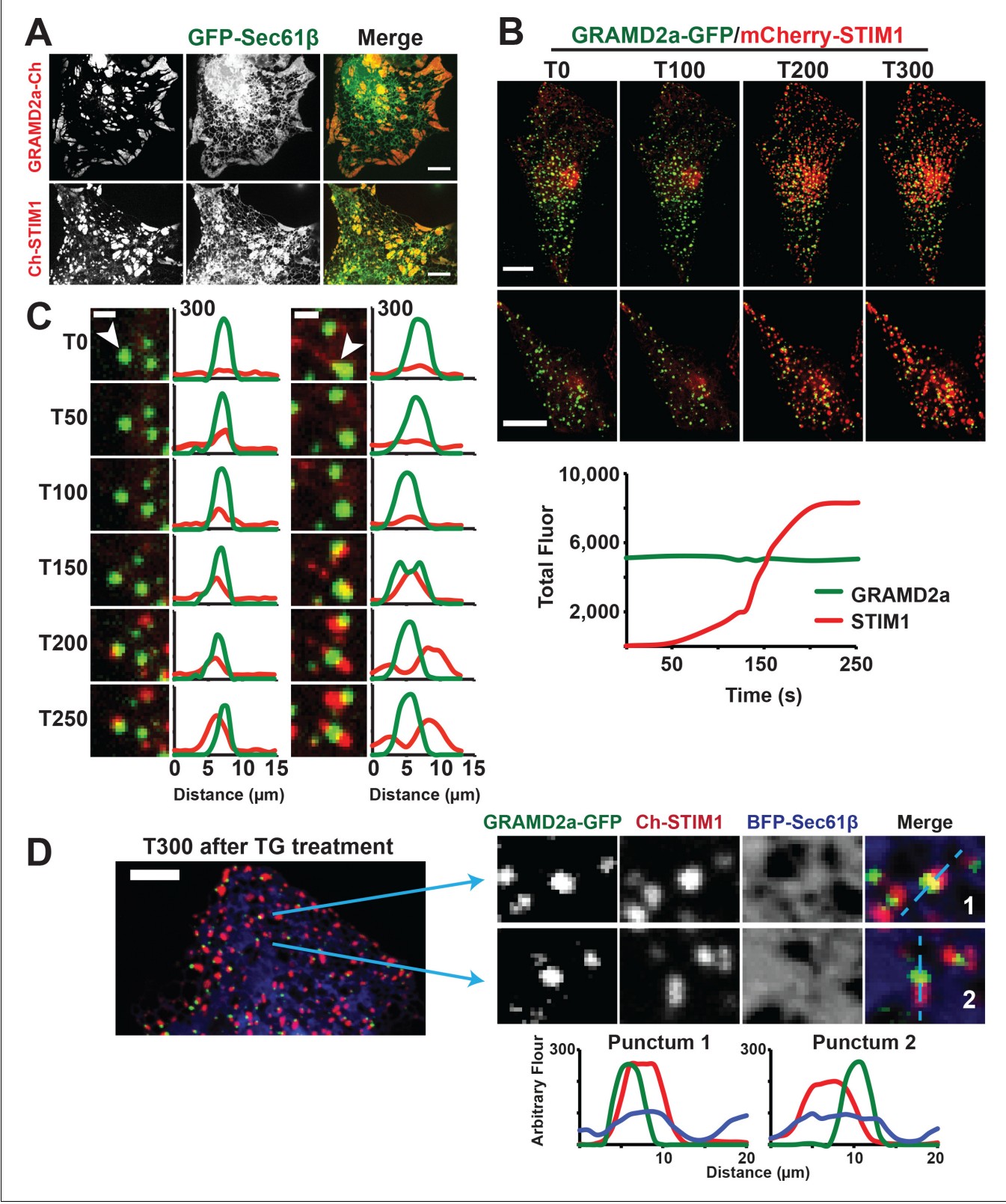

**Figure 5.** GRAMD2a pre-marks ER-PM membrane contact sites used for STIM1 recruitment during SOCE. (**A**) Cortical ER as visualized with ER marker GFP-Sec61β in cells overexpressing GRAMD2a-mCherry (top panel) or mCherry-STIM1 (bottom panel). (**B**) Behavior of mCherry-STIM1 at resting $Ca^{2+}$ and upon depletion of $Ca^{2+}$ stores from the ER using SERCA inhibitor thapsigargin (TG). At T = 30 s, 1 μM TG was added. Two representative cells shown from two different experimental days. Lower panel left shows sample graph of total fluorescence of GRAMD2a-eGFP and mCherry-STIM1

*Figure 5 continued on next page*

*Figure 5 continued*

throughout duration of experiment for a sample cell. (**C**) Zoomed-in images and line scans of individual GRAMD2a fluorescent puncta and associated fluorescence for duration of the TG treatment. Left and right panels are examples of puncta from two different cells. Y-axis of line scans are arbitrary fluorescence units. Scale bar is 2 µm. (**D**) Inset cropped from top cell in (**B**). showing COS7 cell expressing GRAMD2a-eGFP, mCherry-STIM1, and BFP-Sec61β at T300s of the TG-stimulated Ca$^{2+}$ depletion. Corresponding zoomed-in images and line-scans are shown in the bottom panel. Representative images shown from at least 19 cells that were obtained from three biological replicates.

DOI: https://doi.org/10.7554/eLife.31019.016

The following video, source data, and figure supplement are available for figure 5:

**Source data 1.** Quantification of the percentage of co-localized total fluorescent pixels of GRAMD2a-GFP with mCherry-STIM1 or mCherry-STIM1 with GRAMD2a-GFP as a function of time after TG addition.

DOI: https://doi.org/10.7554/eLife.31019.018

**Source data 2.** *Figure 5B* representative line graph: Total pixels of GRAMD2a-eGFP and mCherry-STIM1 fluorescence during TG-treatment experiments (1 µM TG is added at T = 30 s) for sample cell.

DOI: https://doi.org/10.7554/eLife.31019.019

**Figure supplement 1.** Supporting evidence for STIM1 recruitment to GRAMD2a-marked ER-PM contact sites.

DOI: https://doi.org/10.7554/eLife.31019.017

**Figure 5—video 1.** GRAMD2a + STIM1 during TG treatment time-lapse for top cell displayed in *Figure 5B*

DOI: https://doi.org/10.7554/eLife.31019.020

**Figure 5—video 2.** GRAMD2a + STIM1 during TG treatment time-lapse for bottom cell displayed in *Figure 5B*

DOI: https://doi.org/10.7554/eLife.31019.021

---

significantly co-localize with cortical GRAMD1a foci following TG treatment (*Figure 5—figure supplement 1D*). Thus, our observations further support the conclusion that GRAMD1a and GRAMD2a define distinct ER-PM contacts and indicate that GRAMD2a functions as a tether that pre-marks ER-PM contact sites specialized for STIM1 recruitment and SOCE.

Depletion of Ca$^{2+}$ in the ER triggers the re-localization of STIM1 to the PM via an interaction of its C-terminal polybasic tail with PM PI(4,5)P2 (*Liou et al., 2007*; *Wu et al., 2014a*). To further test whether GRAMD2a pre-marks MCSs destined for SOCE, we examined the behavior of a STIM1 mutant lacking the C-terminal PI(4,5)P2 targeting domain (STIM1ΔK) (*Zhou et al., 2013*). TG-induced translocation of STIM1ΔK to the PM has been reported to be strictly dependent on Orai1 overexpression (*Liou et al., 2007*; *Park et al., 2009*). We examined TG-induced STIM1ΔK behavior in HeLa and COS7 cells and observed STIM1ΔK translocation to the PM is cell type dependent, where TG-induced translocation of STIM1ΔK is observed in COS7 cells and not observed in HeLa cells (*Figure 6A* and *Figure 6—figure supplement 1A*). In COS7 cells, in contrast to wild type STIM1, the localization of STIM1ΔK did not significantly overlap at any time with GRAMD2a-eGFP-marked cortical regions (*Figure 6A and B*). This observation indicates that STIM1ΔK is defective for targeting to PI(4,5)P2-enriched ER-PM MCSs marked by GRAMD2a. Indeed, our analysis indicates that STIM1ΔK possessed no specificity as it co-localized to approximately the same extent to GRAMD1a-marked cortical regions (*Figure 6B and C*). Thus our data are consistent with GRAMD2a as a PI(4,5)P2-dependent ER-PM tether that pre-marks specific ER-PM MCSs utilized for SOCE.

## GRAMD2a is required for normal PM STIM1 recruitment

Given that GRAMD2a pre-marks sites of STIM1 localization at ER-PM contacts, we asked if it was required for normal STIM1 recruitment to ER-PM contacts during SOCE. We generated a GRAMD2a knock-out (KO) U2OS cell line using CRISPR genome editing (*Figure 7—figure supplement 1A*) and examined the kinetics of STIM1 recruitment the cortical ER (*Figure 7A* and *Figure 7—figure supplement 1B*). In GRAMD2a KO cells, the kinetics and degree of mCherry-STIM1 accumulation at the cell cortex was significantly reduced as compared to control cells following TG treatment (*Figure 7A*). The defect observed in STIM1 translocation to the PM in GRAMD2a KO cells was fully complimented by transiently expressing GRAMD2a-eGFP in GRAMD2a KO cells, indicating that the defect was a consequence of loss of GRAMD2a function and that GRAMD2a-eGFP is functional (*Figure 7A*). In addition, although the number of mCherry-STIM1 cortical puncta was not significantly different between control and GRAMD2a KO cells, the average area of mCherry-STIM1 cortical puncta was 2-fold lower in GRAMD2a KO cells as compared to control cells (*Figure 7—figure supplement 1B and C*). Decreased STIM1 puncta size in GRAMD2a KO cells was not a consequence of

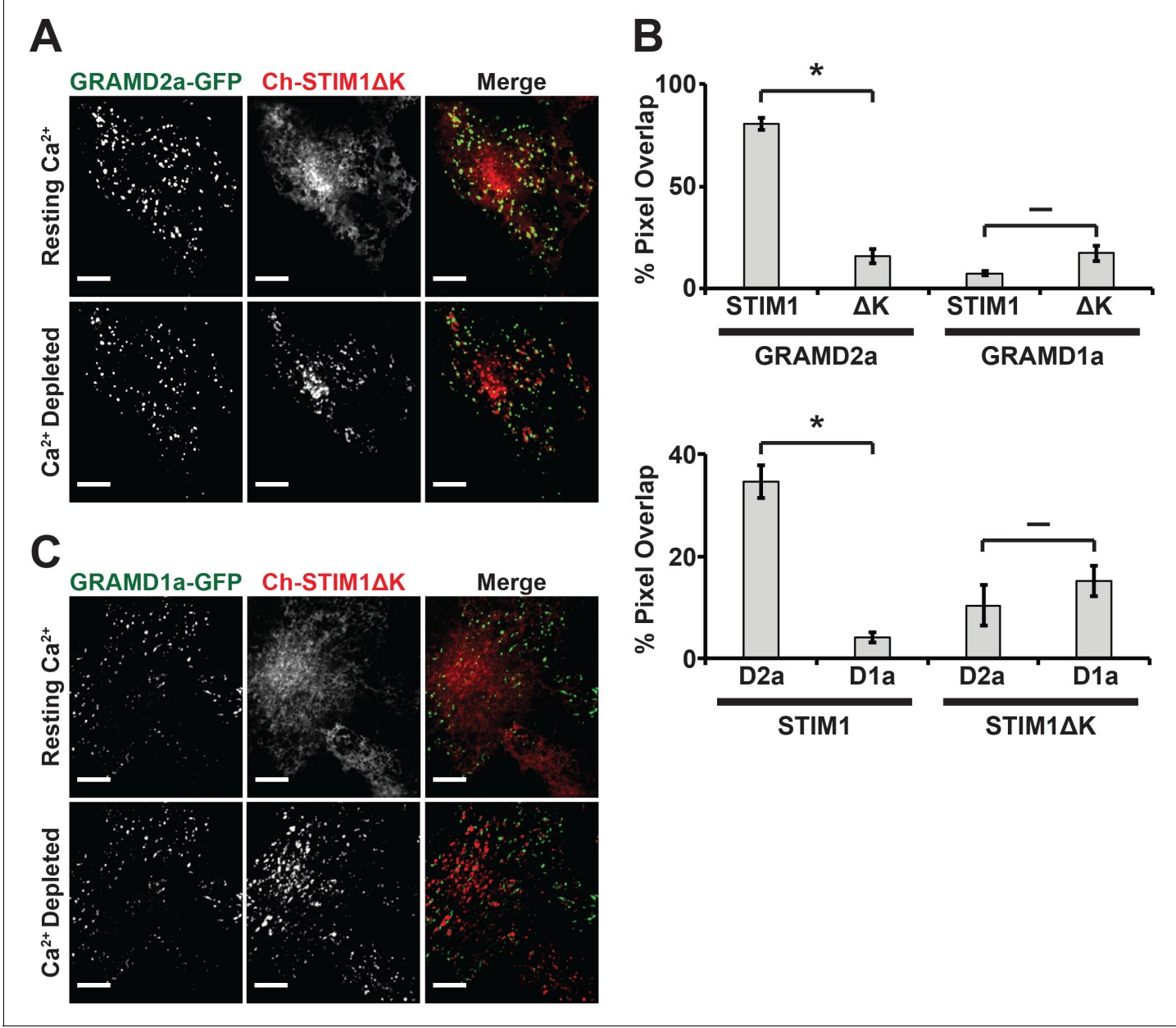

**Figure 6.** STIM1ΔK is defective for targeting to PI(4,5)P2-enriched ER-PM contact sites marked by GRAMD2a. (A–C) Fluorescence images of Cos7 cells expressing mCherry-STIM1ΔK, a PI(4,5)P2-insensitive mutant, and either (A) GRAMD2a-eGFP or (C) GRAMD1a-eGFP before and after TG treatment along with (B) quantification of co-localization. Standard Error shown, *p<$10^{-2}$ and ⁻ p>$10^{-2}$ from two-tailed t-test. Specifically, GRAMD2a with STIM1 compared to GRAMD2a with STIM1ΔK, p=4.47 × $10^{-12}$; GRAMD1a with STIM1 compared to GRAMD1a with STIM1ΔK, p=2.70 × $10^{-2}$; STIM1 with GRAMD2a compared to STIM1 with GRAMD1a, p=5.00 × $10^{-6}$; STIM1ΔK with GRAMD2a compared to STIM1ΔK with GRAMD1a, p=3.64 × $10^{-1}$. Representative images shown from at least 14 cells that were obtained from three biological replicates. Pixel co-localization analysis was performed on all cells.

DOI: https://doi.org/10.7554/eLife.31019.022

The following source data and figure supplement are available for figure 6:

**Source data 1.** *Figure 5B* Bar Graph: Top table is pixel overlap of GRAMD2a and GRAMD1a with STIM1 and STIM1ΔK; standard Error shown.
DOI: https://doi.org/10.7554/eLife.31019.024
**Figure supplement 1.** STIM1ΔK does not respond to TG in HeLa cells.
DOI: https://doi.org/10.7554/eLife.31019.023

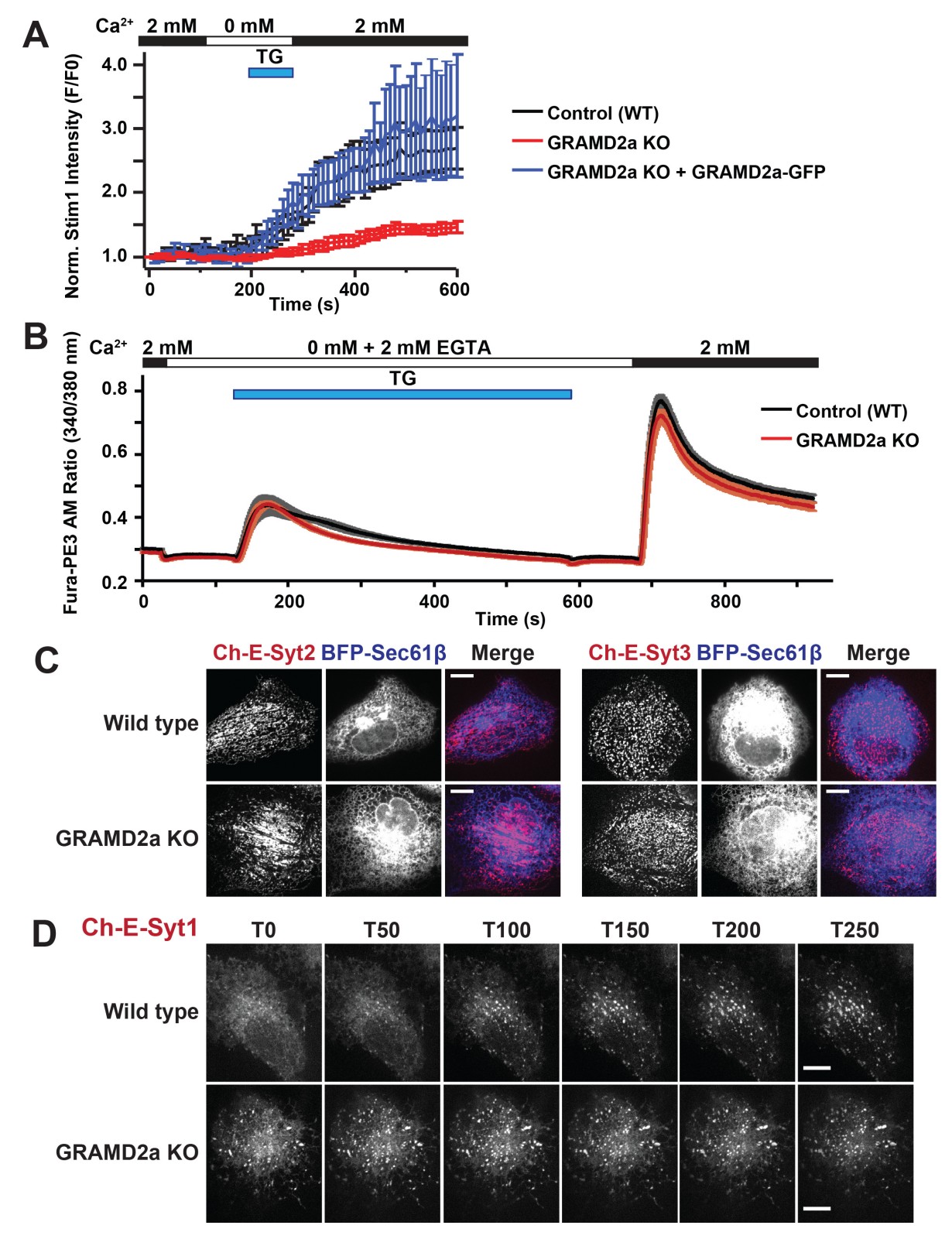

**Figure 7.** GRAMD2a organizes ER-PM domains that selectively function in calcium homeostasis. (**A**) Kinetics and intensity of mCherry-STIM1 recruitment to PM before, during, and after 1 uM TG treatment in wildtype U2OS cells, GRAMD2a knock out (KO) cells, and GRAMD2a KO with GRAMD2a-eGFP transiently transfected. Standard Error is shown, p<0.05 between t = 200 and t-600. (**B**) Cytosolic Ca2+ measurement using ratiomentric dye, Fura-PE3 AM during a Ca2+ addback experiment with wild type U2OS cells and GRAMD2a KO cells. Cells were pre-loaded with 2.5 uM Fura-PE3 AM in 0 mM

*Figure 7 continued on next page*

Figure 7 continued

Ca2+ Ringers and ER stores were loaded in 2 mM Ca2+ Ringers for 4 min prior to recording. Fluorescence was continuously recorded as cells were incubated in the following solutions: (1) Baseline was established in 2 mM Ca2+ ringers for 40 s, (2) cells were washed with 0 mM Ca2+/2 mM EGTA Ringers for 100 s, 3) ER stores were emptied using 4 uM TG in 0 mM Ca2+/2 mM EGTA Ringers for 460 s, 4) cells were returned to with 0 mM Ca2+/2 mM EGTA Ringers for 100 s, (5) cells were washed in 2 mM Ca2+ ringers to observe SOCE across the PM. Comparison of the cytosolic Ca2+ measurements (represented as Fura-PE3 Am ratio) are shown over time. Black trace represented control or wild type U2OS cells; red trace represents GRAMD2a KO U2OS cells. Standard Error is shown, differences between wild type and GRAMD2a traces were not statistically significant as determined using student t-test. nine wild type dishes and 8 GRAMD2a KO dishes were imaged over 2 days. 20–50 cells per dish were imaged. (C) Localization of mCherry-E-Syt2 (left panel) or E-Syt3-mCherry (right panel) in wild type U2OS and GRAMD2a KO cells. (D) Behavior of mCherry-E-Syt1 at resting Ca$^{2+}$ and upon TG-induced ER Ca$^{2+}$ depletion. At T = 30 s, 1 µM TG was added. Top panels show mCherry-E-Syt1 dynamics in wild type U2OS cells and bottom panels show mCherry-E-Syt1 dynamics in GRAMD2a KO cells. For STIM1 and E-Syt1/2/3 experiments, fluorescence intensity dynamics were analyzed for at least 12 cells, where were imaged in biological duplicate.

DOI: https://doi.org/10.7554/eLife.31019.025

The following figure supplements are available for figure 7:

**Figure supplement 1.** STIM1 recruitment is altered in GRAMD2a KO cells.

DOI: https://doi.org/10.7554/eLife.31019.026

**Figure supplement 2.** PM PI(4,5)P2, cholesterol and caveolin-1 are not apparently altered in GRAMD2a KO cells.

DOI: https://doi.org/10.7554/eLife.31019.027

**Figure supplement 3.** E-Syt1 localization is altered in GRAMD2a KO cells.

DOI: https://doi.org/10.7554/eLife.31019.028

decreased expression of endogenous STIM1 protein (*Figure 7—figure supplement 1D*) or differences in the expression level of transfected mCherry-STIM1 protein in wild type and GRAMD2a KO cells as assessed by Western Blot analysis (*Figure 7—figure supplement 1E*, left panel). These data suggest that GRAMD2a functions in the organization of the subset of ER-PM MCSs dedicated to SOCE.

## Loss of GRAMD2a does not affect SOCE, but E-Syt1 localization is significantly altered

Given that loss of GRAMD2a significantly alters STIM1 recruitment to the PM during SOCE, we asked whether Ca2+ homeostasis was also affected in GRAMD2a KO cells during TG-induced SOCE. We examined cytosolic Ca2+ in GRAMD2a KO and control cells using the fluorescent ratio-metric Ca2+-indicator dye Fura-PE3 AM, a leakage-resistant version of Fura-2 (*Chen et al., 2017*; *Chen et al., 2016*). SOCE was interrogated using the standard Ca2+ add back experiment (*Figure 7B*). As previously reported, wild type U2OS cells exhibited a stereotypical SOCE Ca2+ response similar to HeLa and HEK293 cells: upon TG-mediated depletion of ER Ca2+ stores, cytosolic Ca2+ increased above the baseline from Ca2+ released from internal stores and, following the addition of 2 mM Ca2+, increased above the baseline from extracellular Ca2+ influx across the PM (*Figure 7B*, black tracing) (*Chen et al., 2017*; *Chen et al., 2016*; *Kawasaki et al., 2006*). Comparison of the cytosolic Ca2+ measurements between wild type and GRAMD2a KO cells using Fura-PE3 AM indicated that there was no significant difference in the SOCE response (*Figure 7B*, compare black versus red tracings, respectively).

We explored whether additional and/or compensatory changes occurred in GRAMD2a cells by characterizing additional features of the PM in GRAMD2a KO cells. Cortical ER area was not significantly different in wildtype and GRAMD2a KO cells as assessed using TIRF microscopy, indicating that although GRAMD2a is a ER-PM tether, additional ER-PM tethers such as E-Syts, likely function redundantly to maintain ER-PM contacts (*Figure 7—figure supplement 2A*). Given that GRAMD2a binds PI(4)P and PI(4,5)P2, we examined PM PI(4,5)P2 lipids utilizing YFP-PH-PLC$_{\delta 1}$. Line scans and quantification of relative YFP-PH-PLC$_{\delta 1}$ intensity at the PM indicated that there were no apparent differences between wild type and GRAMD2a KO cells (*Figure 7—figure supplement 2B*), indicating that loss of GRAMD2a does not grossly alter PM PI(4,5)P2. We also examined whether loss of GRAMD2a affected PM cholesterol as alterations in PM cholesterol have significant effects on plasma membrane composition, architecture and function (*Chierico et al., 2014*). To examine cholesterol distribution in cells, we utilized the cholesterol biosensor mCherry-D4H (*Maekawa and Fairn, 2015*), which binds cholesterol in the cytosolic leaflet of the PM and organelles and observed

similar localization patterns in wild type and GRAMD2a KO cells (*Figure 7—figure supplement 2C*). In addition, we examined the distribution of Caveolin-1 (Cav1), an essential component of caveolae, which specifically binds PM cholesterol, and observed that Cav1-GFP localization was also similar in wild type and GRAMD2a KO cells (*Figure 7—figure supplement 2D*) (*Tagawa et al., 2005*). Finally, we examined the steady state localization of the E-Syt ER-PM tethers. Steady-state localization of constitutive E-Syt ER-PM tethers, mCherry-E-Syt2 and mCherry-E-Syt3, was not apparently different between wild type and GRAMD2a KO cells (*Figure 7C*). In comparison, the localization of the Ca2+-dependent ER-PM E-Syt tether, E-Syt1, was significantly altered in GRAMD2a KO cells (*Figure 7D*) (*Chang et al., 2013*; *Giordano et al., 2013*; *Idevall-Hagren et al., 2015*). Specifically, in wild type U2OS cells, as previously reported, in nominally zero external Ca2+, mCherry-E-Syt1 localized diffusely in the ER and upon TG-induced SOCE localized to punctate structures at ER-PM contacts (*Figure 7D*, top panel, and *Figure 7—figure supplement 3A*, left panel) (*Chang et al., 2013*; *Idevall-Hagren et al., 2015*). In contrast, in GRAMD2a KO cells in nominally zero external Ca2+, mCherry-E-Syt1 was constitutively localized at ER-PM contacts and this distribution pattern was not altered by TG-treatment (*Figure 7D*, bottom panel, and *Figure 7—figure supplement 3A*, right panel). Western Blot analysis indicated that mCherry-E-Syt1 protein levels were comparable in wild type and GRAMD2a KO cells (*Figure 7—figure supplement 1E*, right panel), indicating this localization difference was not a consequence of different expression levels of E-Syt1 in these cells. The punctate localization pattern of mCherry-E-Syt1 in GRAMD2a KO cells was abolished upon extracellular Ca2+ depletion and chelation with EGTA and mCherry-E-Syt1 was localized throughout the ER, indicating that E-Syt1 localization ER-PM contacts in GRAMD2a KO was dependent on cytosolic Ca2+ (*Figure 7—figure supplement 3B*). This observation suggests the possibility that the constitutive localization of E-Syt1 at ER-PM contacts in GRAMD2a KO cells is a consequence of elevated cytosolic Ca2+ relative to wild type cells. Together with the defect in STIM1 translocation, these observations indicate that loss of GRAMD2a results in a more pleiotropic defect in the organization of ER-PM contacts in cells.

## Discussion

Our results identify GRAMD1a and GRAMD2a as a new class of ER-PM tethers in mammalian cells. GRAMD2a is a simple GRAM domain tether that is targeted to the PM via interaction of its GRAM PH-like domain with PIP lipid determinants – a mechanism shared by a majority of ER-PM tethers (*Figure 8*) (*Chung et al., 2015*; *Heo et al., 2006*; *Raychaudhuri et al., 2006*; *Stefan et al., 2011*). In contrast, GRAMD1a, a tether containing both GRAM and VaST domains, interacts with the PM in a PIP lipid independent manner, raising the possibility that it may have a proteinaceous PM partner and/or be targeted via another PM lipid species, such as cholesterol given the characterized role of the VaST domain in sterol transport (*Gatta et al., 2015*; *Murley et al., 2017*; *Tong et al., 2018*; *Horenkamp et al., 2018*). In this context, the localization of GRAMD1a to ER-PM contacts is cell-type specific; GRAMD1a-marked ER-PM contacts are present in Cos7, Hek293, HeLa, and Arpe19, cells but absent in U2OS or HCT116 cells (*Figure 8—figure supplement 1A*).

Our results conclusively demonstrate the co-existence of physically distinct ER-PM domains at steady state in mammalian cells, uniquely defined by GRAMD1a or GRAMD2a. Consistent with their distinct composition and spatial resolution, GRAMD1a and GRAMD2a-marked ER-PM contact sites also play discrete functional roles at both the cellular and physiological level. The cellular function of GRAMD1a ER-PM contacts sites is currently not known, but the presence of yeast Ltc/Lam-related VaST domain, which mediates inter-membrane sterol transport, suggests GRAMD1a, similar to the yeast Ltc3/4 homologs, may play a role in the regulation of sterol homeostasis (*Gatta et al., 2015*; *Murley et al., 2017*).

Our data show that GRAMD2a functions as a constitutive tether that pre-marks and specifies ER-PM contact sites destined for SOCE, providing an explanation of previous work demonstrating that only a subset of E-Syt2/3-marked ER-PM contacts are utilized by STIM1 (*Figure 8*) (*Carrasco and Meyer, 2011*; *Giordano et al., 2013*; *Várnai et al., 2007*). Consistent with a role for GRAMD2a in SOCE, in its absence we observed that the kinetics and extent of STIM1 translocation to ER-PM contacts during SOCE was significantly altered. However, loss of GRAMD2a did not significantly affect the number of STIM1 puncta or the SOCE Ca2+ response in cells, emphasizing the redundancy of ER-PM tethers, such as E-Syt1/2/3 and oxysterol-binding protein (OSBP)/OSBP-related proteins in

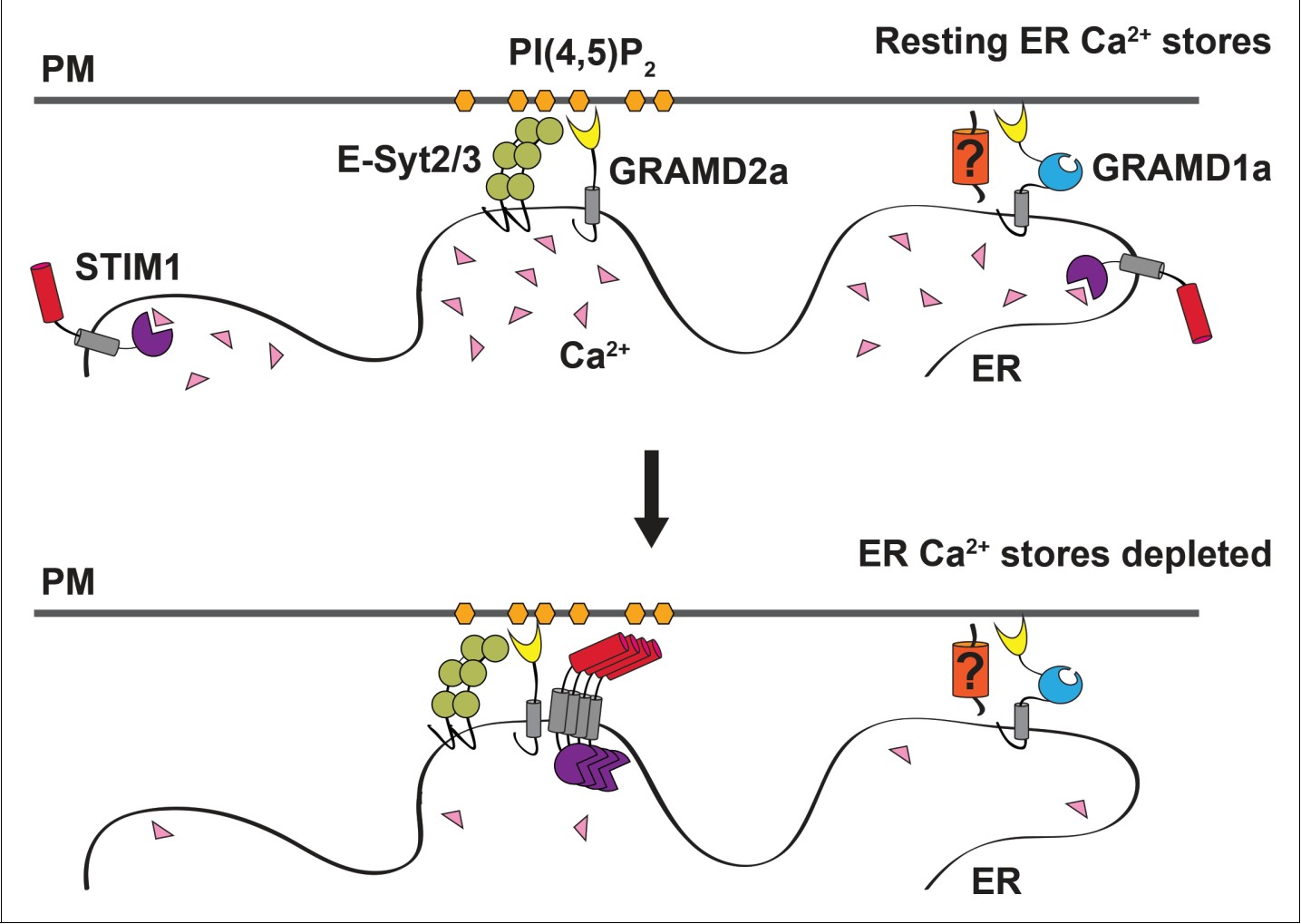

**Figure 8.** Model of GRAMD2a function. GRAMD2a is a constitutive PI(4,5)P2-dependent ER-PM tether that localizes to a subset of E-Syt2/3 contacts. GRAMD1a marks distinct ER-PM contacts in a PI(4,5)P2-independent manner. GRAMD2a pre-marks ER-PM contacts destined for SOCE and facilitates, STIM1 recruitment during SOCE.

DOI: https://doi.org/10.7554/eLife.31019.029

The following figure supplement is available for figure 8:

**Figure supplement 1.** GRAMD1a localization is cell line dependent.

DOI: https://doi.org/10.7554/eLife.31019.030

the maintenance or ER-PM contacts (*Saheki and De Camilli, 2017a*). Although the SOCE Ca2+ response was not apparently affected in cells lacking GRAMD2a, the Ca2+-dependent ER-PM E-Syt1 tether was abnormally localized to ER-PM contacts under conditions in which the internal ER Ca2+ stores were full. Relatively small changes in basal cytosolic Ca2+ are sufficient for E-Syt1 PM recruitment, raising the possibility that loss of GRAMD2a alters cytosolic Ca2+ in a range or spatial manner not detected by Fura-2 (*Idevall-Hagren et al., 2015*). In addition, altered E-Syt1 PM localization in GRAMD2a KO cells suggests that there is a constitutive engagement of E-Syt1's functions at the PM, which include Ca2+-dependent E-Syt1-mediated lipid transport and PI(4,5)P2 maintenance, which, in turn, likely alter cellular physiology, including Ca2+ homeostasis. This raises the possibility of a compensatory Ca2+ homeostatic mechanism at ER-PM contact sites in cells lacking GRAMD2a (*Bian et al., 2018*; *Chang et al., 2013*; *Idevall-Hagren et al., 2015*).

Together with our observation that GRAMD2a pre-marks sites of SOCE, our data suggest a model in which GRAMD2a serves as a constitutive master tether at MCSs with pleiotropic functions including STIM1 recruitment and Ca2+ homeostasis (*Figure 8*). Consistent with this model, GSEA

analysis linked GRAMD2a to a variety of calcium pathways in the cell (*Figure 3B*, lower panel). Mechanistically, given its domain structure and size, GRAMD2a is likely to create a tight ER-PM junction, which may uniquely be required to organize this putative domain and facilitate STIM1 recruitment, consistent with the observed preference of STIM1 for the relatively narrow junction created by E-Syt1 over E-Syt2/3 (*Fernández-Busnadiego et al., 2015*). Our cytological data also indicate that upon ER $Ca^{2+}$ depletion, GRAMD2a and STIM1 initially are co-localized at ER-PM contacts but subsequently become spatially resolved within a contact, suggesting that GRAMD2a may also play a role in the formation of distinct microdomains within ER-PM contacts during SOCE, which would represent an additional level of complexity in the biogenesis and functionality of MCSs. In the absence of GRAMD2a, we speculate that this domain is altered, resulting in compensatory changes in calcium regulatory networks (*Déliot and Constantin, 2015*).

Based on our observations from yeast and mammalian cells, we speculate that members of the Ltc/GRAM superfamily are ER tethers whose partner specificity is coded by their conserved GRAM domain – the defining family feature (*Murley et al., 2015*). The shared ER localization of the Ltc/GRAM proteins in cells raises the possibility that combinatorial intermolecular interactions may occur between family members within the ER to further regulate their localizations and functions beyond ER-PM contacts within the ER-contact site network. As such, our work forms the basis for future experiments aimed at understanding the molecular machinery that regulates cross-talk and interaction between ER and other organelles in a dynamic cellular environment.

# Materials and methods

## Key resources table

| Reagent type (species) or resource | Designation | Source or reference | Identifiers |
|---|---|---|---|
| 6s-His Tag monoclonal antibody | Western Blot antibody | ThermoFisher (His.H8) | RRID:AB_557403 |
| STIM1 monoclonal antibody | Western Blot antibody | ThermoFisher (CDN3H4) | RRID:AB_2197884 |
| GAPDH polyclonal antibody | Western Blot antibody | SigmaAldrich (G9545) | RRID:AB_796208 |
| anti-mCherry polyclonal antibody | Western Blot antibody | ThermoFisher (PA5-34974) | RRID:AB_2552323 |
| Goat anti-mouse or anti-rabbit antibodies | Western Blot antibody (DyLight 800 and DyLight 680) | ThermoFisher | |
| Oxotremorine M | Non-selective muscarinic acetylcholine receptor agonist | SigmaAldrich | |
| Thapsigargin | Non-competitive SERCA inhibitor | Invitrogen/Life-technologies | |
| Fura-PE3 AM | Ratiometric cytosolic calcium indicator | Teflabs | |
| Lipofectamine2000 | Tissue culture transfection reagent | ThermoFisher | |
| Lipids | All lipids used for liposome binding assays | Avanti Lipids | |
| COS7 cells | Cercopithecus aethiops kidney cell line | G. Voeltz (U of Colorado, Boulder) | RRID:CVCL_0224 |
| U2OS cells | Human bone osteosarcoma epithelial cell line | G. Voeltz (U of Colorado, Boulder) | RRID:CVCL_0042 |
| HeLa cells | Homo sapiens cervix adenocarcinoma cell line | P. De Camilli (Yale U) | RRID:CVCL_0058 |
| ARPE19 cells | Human retinal pigment epithelial cell line | ATCC (CRL-2302) | RRID:CVCL_0145 |
| HEK293 cells | Homo sapiens embryonic kidney cell line | E. J. Dickson collection | RRID:CVCL_0045 |
| Fiji (ImageJ) software | Software used to analyse all microscopy images | ImageJ | RRID:SCR_002285 |
| ImageStudioLight | Software used to analyse all Western Blots | LI-COR BioSciences | RRID:SCR_014211 |
| Mega7 | Software used to analyse all Western Blots | MEGA Software | |

## Antibodies and chemicals

6x-His Tag monoclonal primary antibody (His.H8, ThermoFisher), STIM1 monoclonal antibody (CDN3H4, ThermoFisher), mCherry polyclonal antibody (PA5-34974, ThermoFisher), GAPDH polyclonal antibody (G9545, SigmaAldrich), goat anti-mouse and anti-rabbit secondary antibodies DyLight 800 and DyLight 680 (ThermoFisher) were used for western blots. OxotremorineM (SigmaAldrich), thapsigargin (Invitrogen/Life-technologies), isopropylthio-β-galactoside (ThermoFisher), Fura-PE3 AM (Teflabs), Protease Inhibitor (PI) Cocktail Set 1 (Calbiochem/EMD Millipore), 0.01%

poly-L-lysine solution (Sigma), Ni-IDA resin (Protino/Macherey-Nagel), Lipofectamine2000 (Thermo-Fisher). Following concentration of chemicals are used in all the experiments unless noted: oxotre-morineM, 10 μM; thapsigargin, 1 μM; isopropylthio-β-galactoside, 0.5 μM. All lipids were obtained from Avanti Polar Lipids; 1,2-dioleoyl-sn-glycero-3-phosphocholine (DOPC), 850375C; 1-palmitoyl-2-oleoyl-sn-glycero-3-phospho-L-serine 16:0-18:1 (POPS), 840034C; 1,2-dioleoyl-sn-glycero-3-phos-phoserine (DOPS), 840035; L-α-phosphatidylinositol-4,5-bisphosphate (PI(4,5)P2), 840046; L-a-phos-phatidylinositol-4-phosphate (PI(4)P), 840045X; L-a-phosphatidylinositol (PI), 84042C.

## Cell culture and transfection

COS7, U2OS, ARPE19, HEK293 and HeLa cells were cultured in Dulbecco's modified Eagle's medium (DMEM) containing 10% fetal bovine serum (FBS) and 1% penicillin/streptomycin (P/S) at 37°C and 5% $CO_2$. Transfection of plasmids was carried out with Lipofectamine2000 (ThermoFisher) according to manufacturer's instructions. Wild-type as well as genome-edited U2OS cell lines were verified as free of mycoplasma contamination by a PCR-based method. All cell-based experiments were repeated at least two times.

## Fluorescence microscopy

For imaging experiments, cells were plated at low density (120,000 to 150,000 cells/plate or cover-slip) on 35 mm poly-D-lysine glass bottom dishes (MatTek Corp) or on 25 mm circular glass cover-slips (FisherScientific) that were coated with 0.01% poly-L-lysine solution (Sigma). Live cell imaging was carried out one day after transfection using one of the following three microscopes: (1) confocal microscopy was performed using spinning disc module of a Marianas SDC Real-Time 3D Confocal-TIRF microscope (Intelligent Imaging Innovations, 3i) fit with a Yokogawa spinning disk head, a 100 × 1.46 NA objective (Olympus) and EMCCD camera, controlled by Slidebook software (Intelligent Imaging Innovations, 3i); (2) confocal microscopy for all of the OxoM and some TG treatment experiments were performed using Zeiss 880 laser-scanning microscope, equipped with an Airyscan detector; a 63x oil immersion lens, controlled by Zen software (Zeiss); (3) total internal reflection fluorescence (TIRF) microscopy was performed on a Nikon motorized TIRF rig setup built around a Nikon TiE microscope equipped with perfect focus and 100x/1.49 CFI Apo TIRF oil immersion objectives, controlled by Elements software (Nikon). Unless otherwise stated, before imaging cells were transferred to Ringers solution containing 160 mM NaCl, 2.5 mM KCl, 2 mM CaCl2, 1 mM MgCl2, 10 mM Hepes, and 8 mM glucose, pH 7.4 (NaOH). Unless otherwise stated, GRAMD1a-GFP and GRAMD2a-GFP are transfected at 100 ng per 35 mm MatTek dish. In overexpression experiments (*Figure 5A* and *Figure 5—figure supplement 1B*), 1 μg of both GRAMD1a-GFP and GRAMD2a-GFP was transfected.

## GRAMD2a purification

Soluble GRAMD2 without the C-terminal transmembrane domain (GRAMD2ΔTM: 1–298 aa) was cloned into pET15b vector. His$_6$-GRAMD2ΔTM was purified from BL21 *E. coli* containing RIPL plas-mid (encoding nonabundant tRNAs). All expression and purification were performed with cultures were grown to at 37°C. Cultures were grown to OD600 0.7 and expression was induced with 0.5 mM IPTG for 2 hr. Cells were harvested routinely by centrifugation, resuspended in a buffer (50 mM Hepes, pH 8, 500 mM NaCl, 350 μg/ml PMSF, 1x PI cocktail, 2 mM BME) and lysed in a microfluid-izer, upon which Triton X-100 was added to 0.1%. Insoluble proteins and debris were removed by centrifugation at 35,000 rpm in a Beckman 45Ti rotor for 45 min. Column packed with Ni-IDA resin, which was loaded onto a BioRad FPLC, was used to purify His$_6$-GRAMD2ΔTM. Peak fractions were collected, pooled and dialyzed into storage buffer. Recombinant protein stored in 30 uL aliquots at −80°C. Additional buffers used: wash buffer – 50 mM Hepes, pH 8, 500 mM NaCl, 1x PI cocktail, 2 mM BME, 0.1% Mega-8, 5 mM Imidazole; elution buffer – 50 mM Hepes, pH 8, 250 mM NaCl, 1x PI cocktail, 2 mM BME, 0.1% Mega-8, 250 mM Imidazole; dialysis buffer: 50 mM Hepes, pH 8, 150 mM NaCl, 2 mM BME; storage buffer – 50 mM Hepes, pH 8, 150 mM NaCl, 2 mM BME, 10% glycerol.

## Liposome binding assays

Lipids were obtained from Avanti Polar Lipids in chloroform or methanol. Liposomes were prepared as described in *DeVay et al. (2009)*. In short, liposomes were mixed in the appropriate ratios and

dried under nitrogen gas. The dried lipids were placed under vacuum for at least 1 hr, and hydrated in 20 nM Tris-HCl, pH 8, 150 mM NaCl for 20 min using a sonicating bath. Hydrated liposomes were extruded through a 1 µm nanopore membrane (GE Healthcare) a minimum of 15 times. Compositions of liposomes used in this study were as follows. Control PM-like liposomes: 85% PC: 15% PS; PM-like liposomes with PI: 85% PC: 10% PS: 5% PI or 85% PC: 15% PI; PM-like liposomes with PI(4)P: 85% PC: 14% PS: 1% PI(4)P, 85% PC: 10% PS: 5% PI(4)P, or 85% PC: 15% PI(4)P; PM-like liposomes with PI(4,5)P2: 85% PC: 14% PS: 1% PI(4,5)P2, 85% PC: 10% PS: 5% PI(4,5)P2, or 85% PC: 15% PI(4,5)P2.

Liposomes (final concentration 1.2 mg/mL) were incubated with 1.2 µM recombinant GRAMD2ΔTM at room temperature for 90 min. Following incubation, liposomes and recombinant GRAMD2ΔTM were centrifuged in a TLA-100 rotor in Beckman TLA-100 Ultracentrifuge at 40,000 $g$ for 30 min at 4°C. Supernatant and pellet are separated and pellet was resuspended in equivalent volume of 20 mM Tris-HCl, pH 8, 150 mM NaCl as suppernatant. 6x Laemmli was added to pellet and suppernatent and western blots were performed with samples. Anti-His6 antibody was used to detect recombinant His6-GRAMD2ΔTM. Western blots were scanned on LI-COR Odyssey Imager.

## Phylogenetic analysis and sequence alignment

Phylogenetic analysis and structure prediction programs reveal that Ltc1/2/3/4 (Lam6/5/4/2) and GRAMD1a/b/c as well as GRAMD2/3 are orthologs of a family of proteins that is anchored via the presence of the GRAM domain (Figure S1A). MEGA7 software (*Kumar et al., 2016*) was used to create a Maximum Likelihood phylogenetic tree of proteins possessing GRAM domains and PH domains from *Homo sapiens*, *Saccharomyces cerevisiae*, and *Drosophila melanogaster*. A phylogenetic tree that was bootstrapped 1000 times indicated that the GRAMD domains from *H. sapian* proteins GRAMD1a/b/c, GRAMD2, and GRAMD3 share a common ancestor with their *S. cerevisiae* protein orthologs Ltc1/2/3/4 (Lam6/5/4/2). GRAMD4, on the other hand, is not evolutionarily related to GRAMD1a/b/c, GRAMD2, or GRAMD3. MUSCLE alignment (*Edgar, 2004*) of GRAM domains from Ltc1/2/3/4 (Lam6/5/4/2) and GRAMD1a/b/c as well as GRAMD2/3 reveals a high degree of conservation among these nine proteins (*Figure 1B*). Sequences used for GRAM domain homology alignment were the following:

Q96CP6 GRAMD1a 91–158
Q3KR37 GRAMD1b 96–163
Q8IYS0 GRAMD1c 69–136
Q8IUY3 GRAMD2a/GRAMD2 72–139
Q96HH9 GRAMD2b/GRAMD3 110–177
Q08001 LTC1/LAM6 164–231
P43560 LTC2/LAM5 198–265
P38800 LTC3/LAM4 548–617
Q06681 LTC4/LAM2 647–716

## ER Ca$^{2+}$ depletion and PI(4,5)P2 depletion experiments

To monitor store-operated calcium entry (SOCE), cells were cultured and transfected on 35 mm Mat-Tek poly-D-lysine coated dishes (MatTek Corporation) or on 25 mm coverslips (ThermoFisher) precoated with poly-L-lysine (Invitrogen). If cells that were plated on 25 mm cover slips were placed into a perfusion chamber (QCM-825-IPB, Quorum Technologies Inc, Ontario) where cells were incubated in a series of buffers. On day of imaging cells were washed twice and incubated with Ca$^{2+}$-free Ringer's solution (160 mM NaCl, 2.5 mM KCl, 1 mM MgCl2, 10 mM HEPES, 8 mM glucose, pH 7.4 (NaOH) for up to 30 min before imaging. 4 min prior to imaging ER stores were preloaded by incubating the cells in Ringer's solution with 2 mM Ca2+. All experiments were performed at 37°C unless otherwise noted.

## STIM1 (*Figure 5*), STIM1ΔK (*Figure 6*), and E-Syt1 (*Figure 7*) experiments

Different solutions were applied to cell by bath application. Cells were incubated in a series of buffers during which cells were imaged every 5 s: (1) after ER Ca2+ stores were preloaded in 2 mM Ca2 + Ringer's solution, cells were washed twice and Ca$^{2+}$-free Ringer's solution was added to the cells; (2) 1 µM thapsigargin (TG) was added in 0 mM Ca$^{2+}$ Ringer's solution 30 s after start of imaging; (3)

2 mM $Ca^{2+}$ Ringer's solution was added 100 s after TG treatment. Imaging was performed using the Marianas SDC Real-Time 3D Confocal-TIRF microscope every 10 s. Excitation light was provided by 405 nm (for BFP), 488 nm (for eGFP), and 561 nm (for mCherry).

### STIM1 experiments in *Figure 7*
Different solutions were applied to cells using a perfusion system. Cells were incubated in a series of buffers during which cells were imaged every 5 s: (1) 2 mM $Ca^{2+}$-containing Ringer's solution for 100 s; (2) 0 mM $Ca^{2+}$ Ringer's solution for 100 s; (3) 0 mM $Ca^{2+}$ Ringer's solution with 1 µM TG for 100 s; (4) 2 mM $Ca^{2+}$ Ringer's solution. Imaging was performed using a 63x TIRF, 1.49 N.A Olympus objective mounted on an Olympus Cell TIRF system. Excitation light was provided by 488 nm (for eGFP) and 561 nm (for mCherry).

### PI(4,5)P2 depletion experiments in *Figure 4*
Different solutions were applied to cells using a perfusion system. Cells were incubated in a series of buffers during which cells were imaged every 5 s: (1) 2 mM $Ca^{2+}$-containing Ringer's solution; (2) between t = 100 s and t = 120 s, cells were incubated Ringer's solution with 10 µM of oxotremorine; (3) 2 mM $Ca^{2+}$-containing Ringer's solution. Cells were imaged using the Zeiss 880 laser-scanning microscope, equipped with an Airyscan detector, every 10 s. Excitation light was provided by 435 nm (for CFP) and 561 nm (for mCherry).

### Cytosolic Ca2+ experiments in *Figure 7*
Cytosolic Ca2+ was measured in the U2OS cells at 32°C. Cells were loaded with 2.5 uM Fura-PE3 AM, a leakage-resistant version of Fura 2 (Teflabs), at 2 µM in nominally Ca2+-free Ringer's solution, for 45 min, followed by 45 min wash to allow de-esterification. Loading time and concentration were the minimum needed to produce acceptable signal-to-noise ratios, in order to minimize any dye-dependent Ca2+ buffering. ER stores were loaded in 2 mM Ca2+ Ringers for 4 min prior to recording fluorescence. Different solutions were applied to cells using a perfusion system: (1) 2 mM $Ca^{2+}$-containing Ringer's solution for 40 s; (2) 0 mM Ca2+/2 mM EGTA Ringers for 100 s; (3) 0 mM Ca2+/2 mM EGTA Ringers solution with 4 µM TG for 460 s; (4) 0 mM Ca2+/2 mM EGTA Ringers solution for 100 s; (5) 2 mM $Ca^{2+}$ Ringer's solution. Fluorescence was detected at $530 \pm 20$ nm FWHM with a photomultiplier, in response to alternating sequential excitation (10 Hz) at 340 and 380 nm, delivered by a Xe arc lamp through a Cairn Optoscan monochromator. In each experiment, fluorescence signals due to 340 and 380 nm from cells within a fixed size region of interest were demultiplexed and separately recorded. For analysis, the 340/380 fluorescence ratio was formed, with each λ first corrected by subtracting cell-free system background from a same-sized area.

## Plasmids, strains, and cloning
The following reagents were kind gifts: pAc-mCherry-E-Syt2 and pAc-E-Syt3-mCherry from P. De Camilli (Yale University) (*Giordano et al., 2013*); pcDNA3.0-HAII-M1R(dark), pYFP-PH-PLCdelta1, and pCFP-PH-PLCdelta1 from B. Hille (University of Washington) (*Suh et al., 2004*) (*Botelho et al., 2000*); pAc-mCherry-E-Syt1 from J. Liou (University of Texas, Southwestern Medical Center) (*Chang et al., 2013*); pmCherry-D4H from G. Fairn (University of Torono) (*Maekawa and Fairn, 2015*); plyn-mCherry, pAc-GFP-Sec61β, pAc-mCherry-Sec61β, pAc-TagBFP-Sec61β, and pEX-SP-YFP_STIM1(23–685) from G. Voeltz (University of Colorado, Boulder) (*Friedman et al., 2011*; *Liou et al., 2005*). pAc-Cav1-GFP was purchased from Addgene (14433).

Wild type HeLa and E-Syt triple knock-out (TKO) cells were kind gifts from P. De Camilli (Yale University) (*Saheki et al., 2016*). HEK293 cells were taken from the collection of E. J. Dickson (author on this study). COS7 and U2OS cells lines were kind gifts from G. Voeltz (University of Colorado, Boulder). ARPE19 cells were purchased from ATCC (CRL-2302).

cDNA clones of human GRAMD1a and GRAMD2a/GRAMD2 were obtained from Origene (GRAMD1a: RC226692, GRAMD2a: RC220145) and cloned into pAcGFP1-N1 and pAcmCherry-N1 vectors. YFP was replaced with mCherry in pEX-SP-YFP_STIM1 to generate pEX-SP-mCherry_STIM1 using AgeI and NotI restriction cloning; poly-K domain of STIM1 was deleted from pEX-SP-mCherry_STIM1 using Gibson Assembly.

## CRISPR genome editing

CRISPR genome editing was performed using donor vector pX330, a human codon-optimized SpCas9 and chimeric guide RNA expression plasmid, as previously described (*Haeussler et al., 2016*). Briefly, primers coding for six different guide RNA sequences were cloned into pX330 with two overlapping primers each. U2OS cells were co-transfected with these vectors and pAc-GFP vector. Two days post transfection, individual GFP-expressing cells were FACS sorted into 96-well plates and clonal isolates were grown up. PCR-based genotyping was performed on ~80 clones and several candidate clones were identified with mutations in the GRAMD2a ORF. These clones were further analyzed by PCR-based genotyping. Guide 3 resulted in 4-nucleotide deletion in Exon 4 as selected for analysis. GRAMD2a exon 4 guide 3 (5' GTTTAAGGATGTTCCCTTGG AGG) was cloned into pX330 with two overlapping primers (Forward_Exon4_Guide 3–5' CACCGTTTAAGGATGTTCCC TTGG and Reverse_Exon4_Guide 3–5' AAACCCAAGGGAACATCCTTAAAC).

## Statistical analysis

Co-localization analysis: ImageJ was used for analysis of co-localization for fluorescently labeled proteins. Confocal time series were exported to ImageJ, a background subtraction performed, and thresholded. Resulting images were made to look as similar as possible to the original confocal micrographs. Binary masks were created of all fluorescent pixels within thresholded images. Masks of fluorescent proteins were subtracted from each other and % pixel overlap was calculated. For example, in *Figure 2C*, the mask for GRAMD2a-GFP fluorescence was subtracted from the mask for Ch-E-Syt2 and visa versa, which resulted in the co-localization values presented in *Figure 2E*. At least 10 cells were quantified, imaged on at least two separate days (biological duplicates) are quantified. Two-tailed t-tests were performed wherever appropriate to access statistical significance. Percent fluorescence of YFP-PH-PLC$_{\delta 1}$ at PM was determined using the following formula: % Fluorescence at PM = (Total integrated fluorescence density of whole cell − integrated fluorescence density of cytosol) / (Total integrated fluorescence density of whole cell) x 100 using ImageJ plugins. At least 10 cells were quantified, imaged on at least two separate days (biological duplicates) are quantified.

Kinetic analysis: ImageJ was used for analysis of time series kinetics. Briefly, cells were background subtracted and a region of interest (ROI) was drawn around the entire TRIF footprint. Resulting time points were then exported to IGOR Pro Software or GraphPad prism for visualization and quantification of data. At least 6 cells, imaged on at least two different days were analyzed.

Measurement of STIM1 puncta size: Confocal time series were exported to ImageJ, a background subtraction performed, and thresholded. It should be noted that care was taken to ensure the resulting images were similar to the original confocal micrographs. Following this step, thresholded images were converted into binary masks and area of cluster size calculated and a frequency histogram generated using GraphPad Prism. At least 8 cells, imaged on at least two different days were analyzed.

Affinity of GRAMD2ΔTM for liposomes: Western blot scans were analyzed using Image Studio Lite software. In order to standardize quantification, a standard box was created to encompass the largest band and, subsequently, this box was used to quantify the total intensity of each band, after background subtraction. Intensity of total protein in a reaction was calculated by adding the intensity of the pellet band with the supernatant band from each reaction. % binding was calculated by dividing intensity of pellet band by intensity of total protein. Each type of liposome binding experiment was performed with all controls (reaction that contained 1 – no liposomes, 2 – PC/PS liposomes) at least 4 times on four different days.

## Gene set enrichment analysis

Transcript expressions from human liver of 234 male and 193 female individuals were obtained from Gene Expression Omnibus (GEO) under the accession number GSE9588 (*Schadt et al., 2008*). Transcript expressions from mouse livers of 41 male BXD strains were previously described (*Wu et al., 2014b*), and are publicly available from GEO under the accession number GSE60149. For enrichment analysis, genes were ranked based on their Pearson correlation coefficients with the expression levels of *GRAMD1A* and *GRAMD2A*, and Gene Set Enrichment Analysis (GSEA) was performed to find the enriched gene sets co-expressed with *GRAMD1A* and *GRAMD2A* (*Subramanian et al.,*

*2005*). Gene ontology (GO) gene sets were downloaded from the Molecular Signatures Database (MSigDB) (*Subramanian et al., 2005*).

## Acknowledgements

We would like to thank Dr. Michael Paddy in the University of California, Davis (UC Davis) MCB-CBS Imaging Facility for advice with fluorescence microscopy. Also, we would like to thank members of the Nunnari, Bers, and Auwerx laboratories for stimulating discussions. M Besprozvannaya is supported by National Institutes of Health (NIH) Ruth L Kirschstein National Research Service Award (NRSA) Individual Postdoctoral Fellowship F32GM117689; J Nunnari is supported by NIH grants R01GM062942, R01GM097432, and R01GM106019. D Bers is supported by NIH grant HL30077. J Auwerx is supported by grants from the École Polytechnique Fédérale de Lausanne, the Swiss National Science Foundation (31003A-140780), the AgingX program of the Swiss Initiative for Systems Biology (51RTP0-151019), and the NIH (R01AG043930, R01AA016957). J Nunnari and J Auwerx are on the Scientific Advisory Board of Mitobridge, Inc. The authors declare no other competing financial interest.

## Additional information

### Competing interests

Johan Auwerx, Jodi Nunnari: Scientific advisory board, Mitobridge. The other authors declare that no competing interests exist.

### Funding

| Funder | Grant reference number | Author |
| --- | --- | --- |
| National Institutes of Health | F32GM117689 | Marina Besprozvannaya |
| National Institute of General Medical Sciences | R01GM062942 | Jodi Nunnari |
| National Institute of General Medical Sciences | R01GM097432 | Jodi Nunnari |

The funders had no role in study design, data collection and interpretation, or the decision to submit the work for publication.

### Author contributions

Marina Besprozvannaya, Conceptualization, Data curation, Formal analysis, Funding acquisition, Validation, Investigation, Visualization, Methodology, Writing—original draft, Writing—review and editing; Eamonn Dickson, Formal analysis, Investigation, Visualization, Methodology, Writing—review and editing; Hao Li, Formal analysis, Validation, Investigation, Methodology, Writing—review and editing; Kenneth S Ginburg, Formal analysis, Investigation; Donald M Bers, Resources, Funding acquisition; Johan Auwerx, Software, Formal analysis, Funding acquisition; Jodi Nunnari, Conceptualization, Resources, Supervision, Funding acquisition, Writing—original draft, Project administration, Writing—review and editing

### Author ORCIDs

Marina Besprozvannaya (iD) http://orcid.org/0000-0001-5856-4130
Hao Li (iD) http://orcid.org/0000-0001-5677-3377
Jodi Nunnari (iD) http://orcid.org/0000-0002-2249-8730

### Decision letter and Author response

Decision letter https://doi.org/10.7554/eLife.31019.033
Author response https://doi.org/10.7554/eLife.31019.034

## Additional files

### Supplementary files
• Transparent reporting form
DOI: https://doi.org/10.7554/eLife.31019.031

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
