## [Decision Letter]

Thank you for submitting your article "GRAM domain proteins specialize functionally distinct ER-PM contact sites in human cells" for consideration by *eLife*. Your article has been favorably reviewed by three peer reviewers, and the evaluation has been overseen by a Reviewing Editor and Randy Schekman as the Senior Editor. The following individual involved in review of your submission has agreed to reveal his identity: Tobias Meyer (Reviewer #2).

The reviewers have discussed the reviews with one another and the Reviewing Editor has drafted this decision to help you prepare a revised submission.

The study by Besprozvannaya and colleagues describes the discovery of a new group of GRAM-domain containing proteins. This work is important as it shows a clear role, particularly of GRAMD2a, in controlling the localization and translocation of STIM1 proteins. The findings in the study strongly argue that GRAMD2a is part of a core machinery that sets up and regulates Ca^2+^/STIM1 signaling and potentially lipid transport at ER-PM junctions. The authors make the additional interesting observation that GRAMD1a and GRAMD2a localize to different ER-PM membrane contact sites. The authors further demonstrate that GRAMD2a pre-marks PI(4,5)P2-enriched ER-PM membrane contact sites for store-operated calcium entry, while GRAMD1a does not co-localize with PI(4,5)P2 regions.

The manuscript is overall interesting and in principle suitable for publication in *ELife*. Nevertheless, the characterization of GRAMD2a KO needs to be made in some more detail, showing whether basal Ca^2+^ levels in the cytosol and receptor or thapsigargin-triggered Ca^2+^ signaling is altered. Also, a statistical analysis of some of the observations in the experiments in Figure 1–Figure 5 needs to be made more quantitatively. While the authors quantify some images they do not quantify other results and still make quantitative statements about them in the text (e.g. significant). Replicates of the individual experiments are, for the most part, not included, and at one point they state data not shown which should be added in Supplementary figures. There are also a few controls missing. Additionally, the authors do not adhere to the order of the figures in the text, which makes it very difficult for the reader to find the piece of data that the text refers to.

Specific Revisions:

1) GRAMD2a KO cells need more characterization.

a) In Figure 7, the single clone could have reduced amount of STIM1 and rescue by GRAMD2a overexpression could result from expansion of MCSs, so please show images to demonstrate normal looking MCSs and document protein levels.

b) Please measure SOCE responses in the KO cells compared to WT

2) The claim that GRAMD2a facilitates or is required for STIM1 recruitment to contact sites seems premature. Please quantitate the analysis to show that most or all of the recruited STIM1 overlaps with GRAMD2a (using total pixel overlap) and provide proper statistical analysis throughout.

3) The localization patterns in Figure 2 need to be quantitated and additional quantitation is needed throughout (see point 2).

4) Does GRAMD2a bind to the PM via a GRAM domain-PIP-lipid interaction? To address this they need to show that deleting or mutating the GRAM domain inhibits binding.

5) Please show something about how the STIM1 separates itself from GRAMD2a during its accumulation at the PM; i.e., are they really part of the same ER structure? Perhaps coexpressing Sec61 marker would suffice. Also, the criteria for judging that GRAM proteins were localized to MCSs was never explicitly stated. Some EM could help a lot here, though it would likely take more than 2 months. A fluorescent marker (MAPPER, or a luminal ER marker) to mark these sites should be tried.

Points that could probably be handled by discussion in the text:

1) Please offer possible explanation as to why STIM1-deltaK forms puncta in cells not overexpressing Orai, contrary to previous studies. For someone in the field, this result will look very suspicious.

2) They should also discuss why GRAMD2a KO does not reduce number of puncta if it acts as a tether.

Reviewer #1

This study shows that GRAMD1a and GRAMD2a localize to distinct ER-plasma membrane contact sites in human cells. Targeting of GRAMD2a to contact sites is dependent on PIP2 and the cytoplasmic domain of GRAMD2a was found to bind PIPs in vitro. GRAMD1a targeting does not seem to be dependent on PIP2. The study goes on to investigate the role of GRAMD2a in store operated calcium entry (SOCE). It shows that GRAMD2a marks contact sites where STIM1 is recruited during SOCE and it is suggested that GRAMD2a is required for the efficient PM translocation of STIM1. Overall, this is very well done and clearly presented study. However, the claim that GRAMD2a plays a role in STIM1 enrichment at contact sites, which is the most interesting aspect of this work, is not yet convincing.

1) A trivial explanation for the results in Figure 7 could be that STIM1 levels are significantly reduced in GRAMD2a KO cells compared to control cells. This could explain why the average size and intensity of STIM1 puncta is reduced in GRAMD2a KO cells but the total number of puncta is not reduced.

2) Figure 5 shows two examples of STIM1 recruitment to the plasma membrane following TG treatment. In both cases, roughly half of STIM1 puncta form at sites that do not appear to be marked by GRAMD2a. Of course, it is possible that the amount of GRAMD2a at these sites is below the level of detection but it is just as likely that GRAMD2a is not necessary for STIM1 recruitment. Since STIM1 puncta at sites that not do contain GRAMD2a seem about as bright and large as the others, the claim that GRAMD2a facilitates or is required for STIM1 recruitment to contact sites seems premature.

Reviewer #2

The study by Besprozvannaya and colleagues describes the discovery of a new group of GRAM-domain containing proteins. This work is important as it shows a clear role, particularly of GRAMD2a, in controlling the localization and translocation of STIM1 proteins. The findings in the study strongly argue that GRAMD2a is part of a core machinery that sets up and regulates Ca^2+^/STIM1 signaling and potentially lipid transport at ER-PM junctions. The authors make the additional interesting observation that GRAMD1a and GRAMD2a localize to different ER-PM membrane contact sites. The authors further demonstrate that GRAMD2a pre-marks PI(4,5)P2-enriched ER-PM membrane contact sites for store-operated calcium entry, while GRAMD1a does not co-localize with PI(4,5)P2 regions.

The manuscript is overall interesting and in principle suitable for publication in *ELife*. Nevertheless, the characterization of GRAMD2a KO needs to be made in some more detail, showing whether basal Ca^2+^ levels in the cytosol and receptor or thapsigargin-triggered Ca^2+^ signaling is altered. Also, a statistics analysis of some of the observations in the experiments in Figure 1–Figure 5 needs to be made more quantitatively. While the authors quantify some images they do not quantify other results and still make quantitative statements about them in the text (e.g. significant). Replicates of the individual experiments are, for the most part, not included, and at one point they state data not shown which should be added in Supplementary figures. There are also a few controls missing. Additionally, the authors do not adhere to the order of the figures in the text, which makes it very difficult for the reader to find the piece of data that the text refers to.

Major Concerns:

- A main piece of data that should be added is a functional characterization of the GRAMD2a KO. As mentioned above, they should measure both basal Ca^2+^ levels and Ca^2+^ kinetics in KO and control cells.

- The authors state that GRAMD1a-eGFP is observed in focal structures at the periphery of cells in Figure 1. In the images, however, it looks like especially the co-localization of GRAMD1a-eGFP and GRAMD2a-eGFP with Sec61beta-Ch is rather at the center of the ER structures. This is also true for Figure 1 and Figure 1—figure supplement 1. To be able to make this conclusion, the authors need to quantify the co-localization at the center and the periphery separately, analyzing the percentage of GRAMD1a and GRAMD2a with BFP-Sec61beta and Lyn-Ch. After quantification of the images the authors should add in the text more clearly how many repetitions they did (independent experiments/wells/images). The authors do not comment on the fact that the ER maker in Figure 1—figure supplement 1 and 1C looks very different. Are there morphology changes in cells overexpressing GRAMD1a-GFP?

- The authors claim that in Figure 2—figure supplement 1 the co-localization corresponds to regions of cortical ER. To make this point it would be important to include a non-cortical ER co-localization and then quantify both conditions.

- The distances on the line scan are quite large. Some puncta the authors report are 5µm in size according to the data the authors present, is this correct? Is this only in cells that overexpress a lot of the respective proteins? (i.e. in Figure 2).

- The authors state that a similar localization pattern is observed in Figure 2 and in 2C. Such a statement can only be made if the authors include a quantification of the co-localization pattern.

- The levels of GRAMD1a-GFP expression in Figure 2 (and Figure 2—figure supplement 1) are low and not comparable with the levels of expression of GRAMD2a-eGFP. Could that be the reason why no significant co-localization with E-Syt3-Ch is observed?

- In Figure 4 the localization of GRAMD2a-Ch/GRAMD1a-Ch is very different from CFP-PH-PLC. Since E-Syt2/3 are also known to localize at PI(4,5)P2-enriched ER-PM contact sites, it might be useful to add this marker for the co-localization analysis.

- The authors use a liposome binding assay to assess whether GRAMD2a biding of PI(4)P and PI(4,5)P2 is specific. This assay seems to lack a non-binding control protein. The authors may include GRAMD1a-Ch as a control since it is not supposed to bind these lipids (as claimed in Figure 4). Another useful control for in vitro binding comparison could be E-Syt2/3.

- In Figure 4 the assay seems non-specific for 15% pf PI(4)P and PI(4,5)P2. The authors do not comment on that observation. Do they think it is a charge interaction mostly or a specific PIP2 binding interaction?

- The authors state that GRAMD2a binds to the PM via a GRAM domain -PIP-lipid interaction. To make this claim, they need to delete the GRAM domain and show a failure of binding (Figure 4—figure supplement 1).

- The sentence that GRAMD2a uniquely marks SOCE-specific ER-PM MCSs (subsection “GRAMD2a pre-marks ER-PM contacts specialized for SOCE***”***) is not backed by data shown up to this point. The conclusion can only be made later.

- Throughout the manuscript the authors prefer to show representative images of their results rather than a quantification of biological replicates. In Figure 7 the authors do not show any images and only show quantification of the results. To get a better understanding of the results in 7B the authors should include representative images of the quantified area. They also should perform throughout the study quantifications of images if they want to be able to use the term "significant".

Reviewer #3

This is an interesting paper that proposes a role for GRAM-domain proteins in specializing ER-plasma membrane contact sites (MCSs) for store-operated calcium entry (SOCE). The molecular signature of MCSs that regulate SOCE is a topic of wide interest. Previous work by De Camilli's group has shown that the E-Syt proteins act as tethers to stabilize MCSs where the ER Ca^2+^ sensor STIM1 localizes. Surprisingly, E-syt knockdown reduced the number of MCSs without significantly reducing SOCE, suggesting there may be a subset of E-Syt+ sites that are specialized to conduct Ca. This paper proposes that GRAMD2a is a MCS tether that serves this function. The authors show that GRAMD2a colocalizes with E-Syt2/3 at MCSs in a PIP2-dependent way. After store depletion STIM1 is recruited initially to GRAMD2a sites, while GRAMD2a KO is associated with reduced STIM1 recruitment. On this basis they conclude that GRAMD2a pre-marks and specializes a subset of MCSs for SOCE.

Overall, these studies have been carefully done and are well presented. The non-overlapping localization of GRAMD1a and 2a and their different dependencies on PIP2 are clear. However, based on the data presented I am not convinced of the specific role of GRAMD2a in marking sites of SOCE, and have a number of questions related to the main conclusions that need to be addressed.

1) The presence of GRAMD1a and 2a at ER-PM contacts needs to be demonstrated by EM. The conclusion is currently based on TIRF which only has a resolution of 100-200 nm and cannot resolve MCSs. EM is the standard for establishing the presence of specific proteins at ER-PM contacts, and was used in the initial studies of E-Syts and STIM proteins (Giordano et al., 2013; Wu et al., JCB 174:803, 2006). In addition, this would help test the authors' speculation that GRAMD2a recruits STIM1 to MCSs by enforcing a particular geometry on the junction.

2) The authors conclude the GRAM domain mediates the interaction of GRAMD2a with PIPs in the PM, but do not directly test this by mutating or deleting the domain. This should be done, especially as GRAMD1a does not require PIP2 for its localization.

3) GRAMD2a is referred to as an ER-PM tether, but the only direct evidence for this is that its overexpression increases the size of ER-PM contacts (Figure 5). At presumably much lower levels of endogenous expression, does it actually have this function? In the GRAMD2a knockout cells, the number of MCSs is not affected (Figure 7), which does not add support to its function as a tether. A possible complication here is that STIM1 was also overexpressed, which is known to distort ER-PM contacts. Does GRAMD2a KO reduce the number of MCSs when STIM1 is not expressed? MAPPER or a luminal ER marker might be useful here.

4) A major point is that GRAMD2a helps recruit STIM1 to MCSs, but the underlying mechanism is completely unknown. Some additional information is needed, especially since the colocalization of the two proteins declines with time. Does GRAMD2a bind to STIM1, either directly or as part of a complex? Are the two proteins actually part of the same MCS? If so, does the MCS expand after arrival of STIM1, with STIM1 entering the new area? This could be investigated with an ER marker or a marker for ER-PM contacts (e.g., MAPPER). Again, EM would be very helpful here.

5) Figure 7 shows reduced recruitment of STIM1 in a single GRAMD2a KO clone, suggesting that endogenous GRAMD2a helps recruit STIM1 to the MCSs. The only argument against this being a clonal artifact is that the STIM1 response is "rescued" by transient transfection with GRAMD2a-GFP. However, this overexpression may compensate for the low recruitment in the KO by increasing MCS size. The authors need to show that GRAMD2a expression does not alter the size of the MCSs and is thus a true rescue of function.

6) To support a role for GRAMD2a in SOCE, it is essential to compare Ca^2+^ influx in response to store depletion in GRAMD2a WT and KO cells, without overexpressing STIM1. This will provide the strongest test of whether endogenous GRAMD2a is actually essential for SOCE.

7) It is very surprising that Ch-STIM1deltaK forms puncta in response to Ca^2+^ store depletion (Figure 6). Several studies (e.g., Liou et al., 2007; Park et al., 2009) have shown that STIM1-deltaK by itself does not localize to ER-PM contacts; if Orai is also present it binds STIM1-deltaK and the proteins trap each other. It seems highly unlikely that COS7 cells express enough endogenous Orai to account for the puncta, so the question is, what do they represent and are they actually at the PM? A more interpretable experiment would be to coexpress Orai1 so that STIM1-deltaK/Orai complexes can form without the need for PIP2, and ask whether the clusters fail to colocalize with GRAMD2a.

8) To conclude that "GRAMD2a pre-marks sites of STIM1 localization at ER-PM contacts", it is necessary to show that most or all of STIM1 that ends up at the PM is recruited to GRAMD2a-positive sites. Otherwise, STIM1 may be responding to something else which happens to partially colocalize with GRAMD2a. How much STIM1 localizes to puncta that are not associated with GRAMD2a?

[Editors' note: further revisions were requested prior to acceptance, as described below.]

Thank you for submitting your article "GRAM domain proteins specialize functionally distinct ER-PM contact sites in human cells" for consideration by *eLife*. Your article has been reviewed by two peer reviewers, and the evaluation has been overseen by Suzanne Pfeffer as Reviewing Editor and Randy Schekman as the Senior Editor. The following individual involved in review of your submission has agreed to reveal his identity: Tobias Meyer (Reviewer #2).

The reviewers discussed the reviews with one another and the Reviewing Editor drafted this decision in the hopes that you can prepare a revised submission. We hope you will be able to submit the revised version within two months that highlights the stronger aspects of the current story and addresses the comments here from Reviewer #1. We try hard not to support multiple rounds of reviews, but this expert referee's comments seemed serious enough that additional revision is required. Overall, this reviewer was less convinced now that GRAMD2a specifically marks ER-PM MCSs for SOCE, as the paper concludes. Specifically:

1) The first question was whether expression of the mCh-STIM1 (not endogenous STIM1 – subsection “GRAMD2a facilitates STIM1 recruitment during SOCE”, Figure 7—figure supplement 1) was lower in the single GRAMD2a KO clone. It is not possible to judge this from Figure 7 because the data are all normalized to initial fluorescence. STIM1 overexpression increases the area of MCSs and this is seen in Figure 7—figure supplement 1; reduced expression of mCh-STIM1 in the KO clone could potentially explain the decreased STIM1 puncta area. The second question was whether GRAMD1a overexpression could have increased the size of MCSs beyond control size, which might account for the increased recruitment of mCh-STIM1 to the PM. Please address these questions.

2) The new calcium measurements have serious problems and overall the results are not consistent with the evidence that GRAMD2a knockout greatly inhibits STIM1 accumulation at ER-PM contact sites, and do not support the contention that GRAMD2a KO affects Ca^2+^ homeostasis more generally. The authors see a slight decrease in SOCE in GRAMD2a KO cells when measured by GECO-Orai1 (Figure 7) and conclude this is consistent with the inhibition of STIM1 localization. There are a couple of problems here. First, the difference between control and KO is very slight. Given the profound inhibition of STIM1 translocation in 7A, and the fact that SOCE is a highly nonlinear function of STIM1 binding to Orai1, one would expect a nearly complete elimination of SOCE in KO cells. More importantly, overexpressing Orai1 without coexpressing STIM1 actually inhibits SOCE, presumably because an excess of Orai1 "dilutes" the pool of endogenous STIM1 and reduces the STIM1:Orai1 binding stoichiometry below the level needed to open the channel (see Hoover and Lewis, PNAS 108:13299, 2011; Li Z et al. J Biol Chem 282:29448, 2007; Soboloff et al. J Biol Chem 281:20661, 2006). Such a dilution effect would explain why Ca influx is slightly reduced with GECO-Orai1 overexpression, but not when Ca is measured by Lck-GCaMP or fluo-4 (and endogenous Orai and STIM levels are maintained). Thus, the slight inhibition of SOCE with GECO-Orai1 cannot be attributed to the absence of GRAMD2a.

The authors report an increase in SOCE in GRAMD2a KO cells, when Ca^2+^ is measured by Lck-GCaMP or fluo-4 (Figure 7—figure supplement 1), and suggest that GRAMD2a may affect Ca^2+^ homeostasis more generally. However, the measurements of SOCE appear flawed. The classic hallmark of SOCE is that after stores are depleted with TG and Ca^2+^ is readded, the Ca^2+^ level overshoots the baseline. However, in Figure 7 as well as Figure 7—figure supplement 1, the apparent Ca^2+^ level in control cells actually drops below baseline after Ca readdition. In the supplementary figure, the baseline Ca is also not stable, precluding a simple interpretation. Overall, these measurements are not convincing.

We suggest repeating these measurements in control and KO cells using a ratiometric indicator like fura-2, which is much more resistant to artifacts from differential dye loading, bleaching, illumination fluctuations, stray light, etc. Show that the control response is consistent with SOCE. If the KO response is larger than control, then further experiments will have to be done to reconcile the result with the apparent inhibition of STIM1 translocation.

Based on the evidence at this point, the reviewers felt that the conclusion that GRAMD2a really acts as a "master tether" that specifies a subset of E-syt MCSs for SOCE is overstated (Discussion section paragraph two). In the absence of GRAMD2a there are still many STIM1 puncta (Figure 7 and Figure 7—figure supplement 1); apparently, GRAMD2a is not required for STIM1 recruitment, although it may help promote it in some way. The new Ca^2+^ imaging data do not show a strong inhibition of SOCE in the absence of GRAMD2a. Taken together, a role of GRAMD2a in specifying sites for SOCE seems questionable at this stage.

Subsection “GRAMD2a pre-marks ER-PM contacts specialized for SOCE”, final paragraph. Li et al. did not report TG-induced STIM1deltaK PM translocation in the absence of Orai1 overexpression. They overexpressed Orai1-mOrange.

Subsection “ER Ca^2+^950 depletion and PI(4,5)P2 depletion experiments” second paragraph: Ca^2+^ concentrations should be mM, not µM.

Please specify which GECO-Orai1 construct was used (there are two in the Dynes et al. paper). Also, specify which GCaMP was used for the Lck-GCaMP experiments (there are many).

[Editors' note: further revisions were requested prior to acceptance, as described below.]

Thank you for submitting your article "GRAM domain proteins specialize functionally distinct ER-PM contact sites in human cells" for consideration by *eLife*. Your article has been reviewed by one peer reviewer, and the evaluation has been overseen by a Reviewing Editor and Randy Schekman as the Senior Editor. The reviewers have opted to remain anonymous.

The reviewers have discussed the reviews with one another and the Reviewing Editor has drafted this decision to help you prepare a revised submission. While we hate to ask for any additional work after revision, the key reviewer felt very strongly that the conclusions are not supported by the current data and a single experiment has the potential to quickly rectify this issue.

First, the reviewers felt that you have responded in a satisfactory way to points 1 and 2.

Regarding Point 3: The scientific basis for rating the quality as low was Ca^2+^ below baseline even when Ca^2+^ is re-added to WT in Figure 7 left, and the lack of a steady Ca^2+^ baseline at the start of the experiment in Figure 7 right.

In the revision, the authors now conclude that Ca^2+^ homeostasis is somehow different in U2OS cells compared to other cells like HEK or HeLa. This is not justified based on the sparse data that are presented. U2OS cells show very similar TG responses to HEK and HeLa; see Chen et al., Sci. Reports 6:22142, 2016; Supplementary Figure S1B. So this is not a cell type difference, but more likely has something to do with the way the cells have been treated. In the experience of the expert reviewer with many different cell types, basal Ca^2+^ can become elevated simply by plating cells on polylysine-coated coverslips, or flowing solutions past them, and one has to be careful not to stimulate Ca^2+^ signaling through these manipulations. The new fluo-4 data from HEK293 cells look great, but they do not validate the aberrant U2OS responses in Figure 7. Also, the low P values comparing responses in WT vs. KO cells do not indicate that the "data are solid"; they merely indicate the two response are significantly different, but not why they differ. If one set of measurements is flawed, significant differences can occur, but they do not necessarily result from the KO. So this is still a problem that needs to be fixed to enable a meaningful comparison of WT and KO responses.

The new E-syt1 experiment is interesting, and does suggest that Ca^2+^ may be elevated in KO cells in the resting state. A likely explanation for both the TG and E-syt1 results is that the Ca^2+^ stores in the U2OS cells after plating are relatively empty, which prevents TG from releasing any more Ca^2+^ from the ER while stimulating tonic influx through SOCE. This also explains why Ca^2+^ drops inside when Ca^2+^ is removed outside (Figure 7). It is unfortunate the authors did not take the suggestion of recording Ca^2+^ responses using fura-2, which would have indicated any differences in basal Ca^2+^ and allowed them to compare resting Ca^2+^ in WT vs KO in a direct way. (An elevated basal Ca^2+^ is not detectable in Figure 7 because the data are all normalized to the initial fluorescence).

The reviewers believe that the story will elicit substantial interest and further work to determine the underlying mechanisms. However, the SOCE data in U2OS cells cannot yet be interpreted. We suggest the following single experiment to clear this up (and to avoid misconceptions about Ca^2+^ handling in U2OS cells): Please measure the SOCE responses using fura-2, as suggested before. This should not be difficult: it should only take a few days and will go far to clarify the conclusions of this story.

---

## [Author Response]

Specific Revisions:1) GRAMD2a KO cells need more characterization.a) In Figure 7, the single clone could have reduced amount of STIM1 and rescue by GRAMD2a overexpression could result from expansion of MCSs, so please show images to demonstrate normal looking MCSs and document protein levels.

We examined endogenous STIM1 levels using Western analysis with an anti-STIM1 monoclonal antibody in control U2OS and GRAMD2a KO cells. This analysis showed that STIM1 levels are comparable in GRAMD2a KO cells and control wild type cells (Figure 7—figure supplement 1). In addition, the STIM1 translocation experiments, shown in Figure 7, were performed with transfected overexpressed mCherry-STIM1.

We used TIRF microscopy to quantify the total amount of ER-PM contact in U2OS wildtype and GRAMD2a KO cells. Using this analysis, the total area of cortical ER is comparable between control and GRAMD2a KO cells and these data are presented in Figure 7—figure supplement 1.

b) Please measure SOCE responses in the KO cells compared to WT

We measured SOCE-specific calcium influx across the PM using a GECO- Orai1 and observed that there was a small, reproducible and significant reduction in Orai1-specific Ca^2+^ influx in GRAMD2a KO cells as compared to wildtype cells (Figure 7). These findings are consistent with the defective STIM1 translocation observed in GRAMD2a KO cells and with our conclusion that GRAMD2a is a SOCE-specific ER-PM tether.

We also measured total calcium influx over the PM as well as total cytosolic calcium following TG treatment (Figure 7—figure supplement 1). We observed that in contrast to Orai1, there was a significant TG-dependent increase in both total calcium influx over the PM and total cytosolic calcium in GRAMD2a KO cells as compared to control cells. This finding suggests that loss of GRAMD2a affects calcium homeostasis more generally, potentially via altered PIP lipid homeostasis and/or by alterations in the activity of other calcium regulatory networks that impinge on other Ca^2+^ channels, such as TRP. These observations are interesting and indicate a broader role for GRAMD2a; however, a detailed analysis of the mechanisms underlying these phenotypes is beyond the scope of the manuscript.

2) The claim that GRAMD2a facilitates or is required for STIM1 recruitment to contact sites seems premature. Please quantitate the analysis to show that most or all of the recruited STIM1 overlaps with GRAMD2a (using total pixel overlap) and provide proper statistical analysis throughout.

We have performed the requested quantification and it is now presented in a new Table 1 embedded in the revised manuscript. Proper statistical analysis is provided throughout the manuscript.

3) The localization patterns in Figure 2 need to be quantitated and additional quantitation is needed throughout (see point 2).

We performed the two-tailed t-test on localization of GRAMD1a/2a with E- Syt2/3 in Figure 2. Co-localization quantification (total pixel overlap) of GRAMD2a with E-Syt2/3 shown was statistically different from GRAMD1a with E-Sty2/3.

4) Does GRAMD2a bind to the PM via a GRAM domain-PIP-lipid interaction? To address this they need to show that deleting or mutating the GRAM domain inhibits binding.

Our deletional analysis of GRAMD2a and GRAMD1a indicate that their predicted respective GRAM domains are necessary for targeting to the PM (Figure 1), substantiating our conclusion that they are ER-PM tethers that, in the case of GRAMD2a, directly tethers to the PM via PIP lipids.

5) Please show something about how the STIM1 separates itself from GRAMD2a during its accumulation at the PM; i.e., are they really part of the same ER structure? Perhaps coexpressing Sec61 marker would suffice. Also, the criteria for judging that GRAM proteins were localized to MCSs was never explicitly stated. Some EM could help a lot here, though it would likely take more than 2 months. A fluorescent marker (MAPPER, or a luminal ER marker) to mark these sites should be tried.

Line-scan analysis of individual puncta of GRAMD2a indicates that GRAMD2a and STIM1 share a ER-PM contact site subsequent to their spatial resolution at later time points of TG treatment (Figure 5).

Our most stringent data used to conclude that GRAMD2a and GRAMD1a are localized to ER-PM contacts was TIRF analysis, which is a standard analysis PM-linked events. All other data, including line scans demonstrating the co-localization of these components with both PM and ER are also consistent with this conclusion. In our hands, MAPPER expression perturbs/expands the cortical ER and thus is not suitable. EM analysis would indeed be time and resource consuming. We chose to use our resources to address questions that were completely unknown, such as calcium dynamics in GRAMD2a KO cells.

Points that could probably be handled by discussion in the text:1) Please offer possible explanation as to why STIM1-deltaK forms puncta in cells not overexpressing Orai, contrary to previous studies. For someone in the field, this result will look very suspicious.

Many labs studying SOCE work with HeLa or Hek293 cell lines. Our analysis was performed using COS7 and U2OS cells. Our data demonstrate that in COS7 cells STIMΔK is capable of translocating to the PM after ER calcium depletion, while, as already reported in the field, STIM1ΔK does not translocate to the PM under similar conditions in HeLa cells. These data are now included as Figure 6—figure supplement 1. We hypothesize that differences in the endogenous expression of STIM1 and Orai1 protein account for this apparent discrepancy between different cell lines.

2) They should also discuss why GRAMD2a KO does not reduce number of puncta if it acts as a tether.

We have included the following statements in our revised discussion in response:

“Although the kinetics and extent of STIM1 recruitment are altered in the absence of GRAMD2a, the number of STIM1 puncta is not affected in GRAMD2a KO cells. We speculate that this GRAMD2a-independent recruitment of STIM1 is a consequence of functional redundancy of additional independent ER-PM tethers, such as E-Syt_1/2_/3 and oxysterol- binding proteins (OSBP)/OSBP-related proteins (Saheki and De Camilli, 2017a).”

“The basis for the rapid and selective recruitment of STIM1 to GRAMD2a- marked contacts may lie in the geometry of the ER-PM contact site created by the GRAMD2a tether and/or its influence on PIP lipid dynamics/concentration. Specifically, given its small size and simple domain structure, GRAMD2a is likely to create a tight ER-PM junction, which may facilitate STIM1 recruitment, consistent with the observed preference of STIM1 for the relatively narrow junction created by E-Syt1 over E-Syt2/3 (Fernandez-Busnadiego et al., 2015).”

[Editors' note: further revisions were requested prior to acceptance, as described below.]

The reviewers discussed the reviews with one another and the Reviewing Editor drafted this decision in the hopes that you can prepare a revised submission. We hope you will be able to submit the revised version within two months that highlights the stronger aspects of the current story and addresses the comments here from Reviewer #1. We try hard not to support multiple rounds of reviews, but this expert referee's comments seemed serious enough that additional revision is required. Overall, this reviewer was less convinced now that GRAMD2a specifically marks ER-PM MCSs for SOCE, as the paper concludes. Specifically:1) The first question was whether expression of the mCh-STIM1 (not endogenous STIM1 – subsection “GRAMD2a facilitates STIM1 recruitment during SOCE”, Figure 7—figure supplement 1) was lower in the single GRAMD2a KO clone. It is not possible to judge this from Figure 7 because the data are all normalized to initial fluorescence. STIM1 overexpression increases the area of MCSs and this is seen in Figure 7—figure supplement 1; reduced expression of mCh-STIM1 in the KO clone could potentially explain the decreased STIM1 puncta area.

Measuring endogenous STIM1 made a lot of sense to us and in fact there seems to be a slight elevation in the STIM1 levels in the GRAMD2a KO cells. Nevertheless, we performed Western blots on both endogenous and overexpressed mCherry-STIM1 extracts from wild type and GRAMD2a KO cells under conditions where we observed defective STIM1 translocation in the KO cells and did not observe any significant difference in the expression levels of mCherry-STIM1 (Figure 7—figure supplement 1).

The second question was whether GRAMD1a overexpression could have increased the size of MCSs beyond control size, which might account for the increased recruitment of mCh-STIM1 to the PM. Please address these questions.

We selected cells that had apparently normal ER-PM contacts site density and area. This is documented in Figure 1 low expression (0.1 ug/dish) and Figure 5 high expression (1.0 ug). Under both these conditions we observe that GRAMD2a pre-marks contacts where STIM1 is recruited in cells. Also, Figure 5 documents that under resting conditions with GRAMD2a low expression (low expression was used in all experiments with the exception of Figure 5) STIM1 is diffusely localized to the ER, indicating that overexpression of GRAMD2a is not sufficient for STIM1 recruitment.

2) The new calcium measurements have serious problems and overall the results are not consistent with the evidence that GRAMD2a knockout greatly inhibits STIM1 accumulation at ER-PM contact sites, and do not support the contention that GRAMD2a KO affects Ca^2+^ homeostasis more generally. The authors see a slight decrease in SOCE in GRAMD2a KO cells when measured by GECO-Orai1 (Figure 7) and conclude this is consistent with the inhibition of STIM1 localization. There are a couple of problems here. First, the difference between control and KO is very slight. Given the profound inhibition of STIM1 translocation in 7A, and the fact that SOCE is a highly nonlinear function of STIM1 binding to Orai1, one would expect a nearly complete elimination of SOCE in KO cells. More importantly, overexpressing Orai1 without coexpressing STIM1 actually inhibits SOCE, presumably because an excess of Orai1 "dilutes" the pool of endogenous STIM1 and reduces the STIM1:Orai1 binding stoichiometry below the level needed to open the channel (see Hoover and Lewis, PNAS 108:13299, 2011; Li Z et al. J Biol Chem 282:29448, 2007; Soboloff et al. J Biol Chem 281:20661, 2006). Such a dilution effect would explain why Ca influx is slightly reduced with GECO-Orai1 overexpression, but not when Ca is measured by Lck-GCaMP or fluo-4 (and endogenous Orai and STIM levels are maintained). Thus, the slight inhibition of SOCE with GECO-Orai1 cannot be attributed to the absence of GRAMD2a.[…]Based on the evidence at this point, the reviewers felt that the conclusion that GRAMD2a really acts as a "master tether" that specifies a subset of E-syt MCSs for SOCE is overstated (Discussion section paragraph two). In the absence of GRAMD2a there are still many STIM1 puncta (Figure 7 and Figure 7—figure supplement 1); apparently, GRAMD2a is not required for STIM1 recruitment, although it may help promote it in some way. The new Ca^2+^ imaging data do not show a strong inhibition of SOCE in the absence of GRAMD2a. Taken together, a role of GRAMD2a in specifying sites for SOCE seems questionable at this stage.

To address concerns that the system we used for measuring Ca^2+^ is somehow fundamentally flawed, we measured the SOCE response in the HEK293 cell line, in which the conical SOCE has been defined. As shown in Figure 7—figure supplement 1F, we can reproduce the stereotypical SOCE response in this line using our experimental conditions, which indicates that the trivial explanations provided above do not account for the different response observed in U2OS cells. As expressed in our rebuttal letter dated November 12th above, we are also willing to share all of our raw data, which is compiled in Figure 7 and Figure 7—figure supplement 1F. In the raw movies, it is apparent that there are no “bleaching, focus drift, movement, etc” issues. Also, the Ca^2+^ responses between wild type and GRAMD2a KO cells are significantly different (p values 10-5)-see Figure 7 legend for details, which indicates that our data are “solid”. We have also added a new experiment examining E-Syt1 localization, which represents an independent approach to test the validity of our Ca^2+^ measurement data. It has been documented that under resting conditions E-Syt1 is localized diffusely in the ER but upon elevated cytosolic Ca^2+^ (independent of SOCE) it translocates to ER-PM contacts (Idevall-Hagren et al. 2105). Figure 7 and Figure 7—figure supplement 1G shows that under resting conditions, in contrast to wild type cells, E-Syt1 is constitutively localized at ER-PM contacts in GRAMD2a KO cells in a Ca^2+^ dependent manner, further substantiating our Ca^2+^ measurements and indicating that loss of GRAMD2a significantly alters the composition of ER-PM contacts.

We removed the GECO-Orai data because of the complication in interpretation resulting from overexpression of Orai. As stated above, we have performed additional experiments to address the specificity of the STIM1 and Ca^2+^ phenotypes in GRAMD2a knockout cells. We observe an additional defect in E-Syt1 localization, consistent with abberant Ca^2+^ homeostasis but we did not observe any apparent differences in PM PIP or cholesterol lipids or in E-Syt2/3 localization. Together we feel these data are consistent with a model in which GRAD2a functions as a ER-PM tether that organizes a PM domain devoted to Ca^2+^ handling.

Subsection “GRAMD2a pre-marks ER-PM contacts specialized for SOCE”, final paragraph. Li et al. did not report TG-induced STIM1deltaK PM translocation in the absence of Orai1 overexpression. They overexpressed Orai1-mOrange.

We have removed the citation and note that in HeLa cells where most of the STIM1deltaK work has been performed, we do not observe TG-induced translocation in the absence of overexpressed Orai, consistent with published work. However, we do observe in COS7 cells (Figure 6 and Figure 6—figure supplement 1). Thus, like the SOCE response in U2OS cells, our data indicate that different cell types are indeed different.

Subsection “ER Ca^2+^950 depletion and PI(4,5)P2 depletion experiments” second paragraph: Ca^2+^ concentrations should be mM, not µM.

Done.

Please specify which GECO-Orai1 construct was used (there are two in the Dynes et al. paper). Also, specify which GCaMP was used for the Lck-GCaMP experiments (there are many).

GECO-Orai data have been removed. We have clarified that we used Lck-GCaMP3G.

[Editors' note: further revisions were requested prior to acceptance, as described below.]

First, the reviewers felt that you have responded in a satisfactory way to points 1 and 2.Regarding Point 3: The scientific basis for rating the quality as low was Ca^2+^ below baseline even when Ca^2+^ is re-added to WT in Figure 7 left, and the lack of a steady Ca^2+^ baseline at the start of the experiment in Figure 7 right.In the revision, the authors now conclude that Ca^2+^ homeostasis is somehow different in U2OS cells compared to other cells like HEK or HeLa. This is not justified based on the sparse data that are presented. U2OS cells show very similar TG responses to HEK and HeLa; see Chen et al., Sci. Reports 6:22142, 2016; Supplementary Figure S1B. So this is not a cell type difference, but more likely has something to do with the way the cells have been treated. In the experience of the expert reviewer with many different cell types, basal Ca^2+^ can become elevated simply by plating cells on polylysine-coated coverslips, or flowing solutions past them, and one has to be careful not to stimulate Ca^2+^ signaling through these manipulations. The new fluo-4 data from HEK293 cells look great, but they do not validate the aberrant U2OS responses in Figure 7. Also, the low P values comparing responses in WT vs. KO cells do not indicate that the "data are solid"; they merely indicate the two response are significantly different, but not why they differ. If one set of measurements is flawed, significant differences can occur, but they do not necessarily result from the KO. So this is still a problem that needs to be fixed to enable a meaningful comparison of WT and KO responses.

We thank the expert reviewer for this full explanation and for pointing out published SOCE data in U2OS cells. Based on these comments and data, we recruited a collaborator to help us conduct the Fura-2 measurements as we do not have the appropriate instrumentation. In the course of the Fura-2 experiments, we determined that Fluo-4 and Llk-GcAMP3G data were indeed flawed as a consequence of technical issues and have been removed. As you will see in revised Figure 7, using Fura-2, we did not observe a significant change in the SOCE response between WT and GRAMD2a KO cells.

The new E-syt1 experiment is interesting, and does suggest that Ca^2+^ may be elevated in KO cells in the resting state. A likely explanation for both the TG and E-syt1 results is that the Ca^2+^ stores in the U2OS cells after plating are relatively empty, which prevents TG from releasing any more Ca^2+^ from the ER while stimulating tonic influx through SOCE. This also explains why Ca^2+^ drops inside when Ca^2+^ is removed outside (Figure 7). It is unfortunate the authors did not take the suggestion of recording Ca^2+^ responses using fura-2, which would have indicated any differences in basal Ca^2+^ and allowed them to compare resting Ca^2+^ in WT vs KO in a direct way. (An elevated basal Ca^2+^ is not detectable in Figure 7 because the data are all normalized to the initial fluorescence).

Our Fura-2 data (not normalized to initial fluorescence – only non-dye-loaded cell background subtracted) suggest that basal cytoplasmic Ca^2+^ is not significantly different between the WT and KO cells. The increased constitutive Ca^2+^-dependent ESyt1 PM localization in GRAMD2a KO cells suggest that Ca^2+^ may be altered proximal to the PM. This aberrant localization is likely a compensatory response to altered ER-PM contacts and may normalize Ca^2+^ homeostasis. At this point, we are hesitant to revisit Llk-GcAMP3G experiments given the general concern that the probe may not be a neutral reporter. Thus, the exact basis of the STIM1 and E-Syt-1 localization defects in GRAMD2a KO cells and their relationship to Ca^2+^ (and π (4,5)P2) homeostasis will require additional experimentation beyond the scope of the manuscript.